# Pyrogallol-Phloroglucinol-6,6-Bieckol from *Ecklonia cava* Attenuates Tubular Epithelial Cell (TCMK-1) Death in Hypoxia/Reoxygenation Injury

**DOI:** 10.3390/md17110602

**Published:** 2019-10-24

**Authors:** Myeongjoo Son, Seyeon Oh, Chang Hu Choi, Kook Yang Park, Kuk Hui Son, Kyunghee Byun

**Affiliations:** 1Department of Anatomy & Cell Biology, Gachon University College of Medicine, Incheon 21936, Korea; mjson@gachon.ac.kr; 2Functional Cellular Networks Laboratory, College of Medicine, Department of Medicine, Graduate School and Lee Gil Ya Cancer and Diabetes Institute, Gachon University, Incheon 21999, Korea; seyeon8965@gachon.ac.kr; 3Department of Thoracic and Cardiovascular Surgery, Gachon University Gil Medical Center, Gachon University, Incheon 21565, Korea; cch624@gilhospital.com (C.H.C.); kkyypark@gilhospital.com (K.Y.P.)

**Keywords:** kidney, ischemia-reperfusion injury, *Ecklonia cava*, phlorotannins

## Abstract

The hypoxia/reoxygenation (H/R) injury causes serious complications after the blood supply to the kidney is stopped during surgery. The main mechanism of I/R injury is the release of high-mobility group protein B1 (HMGB1) from injured tubular epithelial cells (TEC, TCMK-1 cell), which triggers TLR4 or RAGE signaling, leading to cell death. We evaluated whether the extracts of *Ecklonia cava (E. cava)* would attenuate TEC death induced by H/R injury. We also evaluated which phlorotannin—dieckol (DK), phlorofucofuroeckol A (PFFA), pyrogallol phloroglucinol-6,6-bieckol (PPB), or 2,7-phloroglucinol-6,6-bieckol (PHB)—would have the most potent effect in the context of H/R injury. We used for pre-hypoxia treatment, in which the phlorotannins from *E. cava* extracts were added before the onset of hypoxia, and a post- hypoxia treatment, in which the phlorotannins were added before the start of reperfusion. PPB most effectively reduced HMGB1 release and the expression of TLR4 and RAGE induced by H/R injury in both pre- and post-hypoxia treatment. PPB also most effectively inhibited the expression of NF-kB and release of the inflammatory cytokines TNF-α and IL-6 in both models. PPB most effectively inhibited cell death and expression of cell death signaling molecules such as Erk/pErk, JNK/pJNK, and p38/pp38. These results suggest that PPB blocks the HGMB1–TLR4/RAGE signaling pathway and decreases TEC death induced by H/R and that PPB can be a novel target for renal H/R injury therapy.

## 1. Introduction

Ischemia/reperfusion (I/R) injury, which occurs when blood supply to tissues or organs is restored after ischemia, leads to more tissue damage than ischemia itself by enhancing the inflammatory reaction in the reperfused tissue [1].

Renal I/R injury is a major pathophysiology of acute kidney injury (AKI) and can induce AKI after kidney transplantation, partial nephrectomy, renal artery angioplasty, aortic aneurysm surgery, and elective urological surgery when blood supply to the kidney is stopped or decreased during surgery [2]. In addition to AKI, I/R injury leads to the loss of tubular epithelial cell (TEC) function, resulting in delayed graft function and acute or chronic rejection of the transplanted kidney [3]. To decrease the incidence of AKI after surgeries accompanied by I/R injury, treatments to decrease I/R injury should be developed.

High-mobility group protein B1 (HMGB1) is released from injured renal cells and activates Toll-like receptors (TLR), which trigger production of pro-inflammatory cytokines such as tumor necrosis factor-α (TNF-α) and transcription of nuclear factor kappa B (NF-κB) [4,5,6], finally leading to tissue injury after I/R injury in the kidney [7,8]. In I/R injury, TECs play a dual role as both injury initiators (by releasing HMGB1) and targets [9]. Receptor for advanced glycation end products (RAGE) also initiates pro-inflammatory signaling in I/R injury by binding HMGB1 in the liver and heart [10,11].

*Ecklonia cava* is a brown alga that contains phlorotannins, polyphenolic compounds that have multiple biological activities including anti-inflammatory [12,13] and antioxidant activities [14]. One study has shown that polyphenol extract from *E. cava* attenuated renal inflammation induced by a high-fat diet by decreasing pro-inflammatory signaling via TNF-α and NF-κB [15]. However, the effect of *E. cava* on I/R injury has not been studied. 

Here, we evaluated whether phlorotannins from *E. cava* extract would attenuate TEC death induced by I/R injury. Commonly, renal hypoxia/reoxygenation (H/R) was established to simulate renal I/R injury in vitro [16,17] and we also used the H/R model. We used two treatment models: a pre-hypoxia model, in which the phlorotannins were added before the onset of hypoxia, and a post- hypoxia model, in which the phlorotannins were added before the start of reoxygenation. In addition, we evaluated which phlorotannin—dieckol (DK), phlorofucofuroeckol A (PFFA), pyrogallol phloroglucinol-6,6-bieckol (PPB), or 2,7-phloroglucinol-6,6-bieckol (PHB)—would have the most potent effect in the context of H/R injury.

## 2. Results and Discussion 

### 2.1. Attenuation of HMGB1 Release from TECs after H/R Injury by the Phlorotannins from E. cava Extracts

In this study, we used mouse kidney tubular cells (TCMK-1) as TECs. In the pre-hypoxia model, the HMGB1 level was increased by H/R injury in both TEC lysate and supernatant (Figure 1A,B), suggesting that TECs injured by H/R increased the synthesis and release of HMGB1. HMGB1 levels in both the TEC lysate and supernatant were decreased by individual phlorotannins added before TECs were exposed to hypoxia. Among individual phlorotannins, PPB and DK showed the strongest attenuation effects.

In the post-hypoxia treatment model, the HMGB1 level increased by H/R injury was also decreased by adding *E. cava* extracts before reperfusion in both TEC lysate and supernatant. Among the 4 phlorotannins, the effect of DK and PPB was the most significant (Figure 1C,D).

HMGB1 is passively released in response to inflammatory stress or necrosis [18]. Our results showed that the HMGB1 release from TECs was increased after H/R injury and that this increase was attenuated by PPB most significantly among the 4 phlorotannins from *E. cava*.

### 2.2. TLR4 and RAGE Expression Induced by H/R Injury Is Attenuated Most Efficiently by PPB

TLR4 and RAGE expression in TECs induced by H/R injury were attenuated by the 4 phlorotannins in the pre-hypoxia treatment model, among them, PPB had the strongest effect on TLR4 and RAGE expression (Figure 2A–C,E). The patterns were the same in the post-hypoxia treatment (Figure 2A,B,D,F).

TLR4 [19] and RAGE [20] are the primary cell membrane receptors that bind HMGB1. In H/R injury, this binding leads to pro-inflammatory signaling pathway activation by TLR and RAGE.

### 2.3. Expression of NF-kB and Pro-Inflammatory Cytokines after H/R Injury Is Attenuated Efficiently by PPB 

The level of NF-κB was increased in both the pre-hypoxia and post-hypoxia treatment models and was decreased by the 4 phlorotannins, with PPB showing the strong effect (Figure 3). The levels of TNF-α and IL-6 were also increased by H/R injury and were attenuated by adding the 4 phlorotannins, among which PPB showed a strong effect.

During H/R injury, TLR4 initiates the inflammatory response by increasing the production of NF-κB-dependent cytokines such as TNF-α [5,6]. In addition, IL-6 is released upon TLR4 activation during H/R injury and amplifies inflammation [11,21]. The RAGE pathway also induces NF-κB activation and leads to an increase in TNF-α and IL-6 levels [22]. TNF-α and IL-6 induce tubular cell death during renal H/R injury [23].

### 2.4. TEC Apoptosis Induced by H/R Injury Is Attenuated Efficiently by PPB

TEC apoptosis induced by H/R injury was attenuated by adding the phlorotannin from *E. cava* extracts in both the pre-hypoxia and post-hypoxia treatments (Figure 4A–C).

### 2.5. PPB Attenuated Cell Death Signals Induced by H/R Injury

The level of SAPK/JNK was increased in both the pre-hypoxia and post-hypoxia treatment models and was decreased by the 4 phlorotannins, with PPB showing the effect (Figure 4D–G). The extracellular signal-regulated kinases-1 and -2 (Erk1/2), the c-Jun N-terminal kinases (SAPK/JNK), and p38 mitogen-activated protein kinases (MAPKs) are involved in cell death induced by H/R injury [24]. ERK signaling is involved in injury and apoptosis of kidney cells during H/R injury [25,26,27]. Both p38 and JNK pathways induce tubular cell death and their inhibition reduces apoptosis and inflammation induced by H/R injury [28,29]. 

In our study, these cell death signals were increased by H/R injury in TEC and were attenuated most significantly by PPB in both the pre-hypoxia and post-hypoxia treatment models. 

Previous one study shows that intraperitoneally administration of *E. cava* polyphenols at 10 mg/kg and 50 mg/kg decreased rat brain infarct size and neuronal cell apoptosis [30] and the other studies shows that 10 μg/mL or 1–50 µM only DK from *E. cava* have protective effects on oxidative stress-induced apoptosis in endothelial progenitor cells (EPCs) [31], primary cortical neurons and HT22 neurons [32]. However, this study tried to mimic renal ischemia reperfusion injury in vitro and validated effects on 4 phlorotannins including dieckol (DK), phlorofucofuroeckol A (PFFA), pyrogallol phloroglucinol-6,6-bieckol (PPB), or 2,7-phloroglucinol-6,6-bieckol (PHB) from *E. cava*. 

## 3. Materials and Methods

### 3.1. Cell Culture

Mouse renal tubular epithelial cells, TCMK-1, were purchased from Korean Cell Line Bank (Seoul, Korea) and cultured in Dulbecco’s Modified Eagle’s medium (DMEM; Gibco, Waltham, MA, USA) containing 10% fetal bovine serum (FBS; Gibco) and 1% penicillin/streptomycin (Gibco) in a 5% CO_2_ incubator (Thermo Fisher Scientific; Waltham, MA, USA) at 37 °C. Growth medium was changed every 2 days.

### 3.2. Hypoxia/Reoxygenation (H/R) Injury Cell Models

To examine the inhibitory effects of 4 phlorotannins DK, PHB, PFFA, and PPB from *E. cava* extract, we designed two types of H/R cell models using TCMK-1 cells, the experimental method and image scheme is shown in Appendix A.

#### 3.2.1. Pre-Hypoxia Treatment Model

Four prepared phlorotannins (2.5 μg/mL) were added to the growth medium, which was used to treat TCMK-1 cells for 6 h before hypoxia. After treatment, the medium was completely removed, and the cells were washed thoroughly with PBS. Hypoxia medium contained 0.5% FBS and 1% penicillin/streptomycin in DMEM. The TCMK-1 cells were cultured in hypoxia medium with the 4 phlorotannins and exposed to gas mixture of 1% O_2_, 5% CO_2_, 94% N_2_ for 14 h within hypoxia incubator chamber (Stemcell technologies, Cambridge, MA, USA). After flushing the hypoxia chamber with the desired gas mixture, the chamber seals tightly with clamp. After incubation, the hypoxia chamber was dismantled and the hypoxia medium was removed, and the cells were washed with PBS. The cells were cultured with fresh oxygenated medium containing FBS, 1% penicillin/streptomycin, 4 phlorotannins in a mixture of 20% O_2_, 5% CO_2_, 75% N_2_ for 9 h, and then used for experiments. The experimental method and image scheme is shown in Appendix A.

#### 3.2.2. Post-Hypoxia Treatment Model

TCMK-1 cells were cultured in growth medium for 6 h before hypoxia, growth medium was completely removed, and the cells were washed with PBS. Hypoxia exposure was performed following in Section 3.2.1 ‘*Pre-hypoxia treatment model*’ of material and methods section. After the hypoxia treatment, the hypoxia chamber was dismantled and the hypoxia medium was removed, and the cells were washed with PBS. The cells were cultured with fresh oxygenated medium containing FBS, 1% penicillin/streptomycin, 4 phlorotannins in a mixture of 20% O_2_, 5% CO_2_, 75% N_2_ for 9 h, and then used for experiments. The experimental method and image scheme is shown in Appendix A.

### 3.3. Immunostaining

TCMK-1 cells were seeded in an 8-well chamber slide (1 × 10^4^ per chamber well) and incubated for 24 h. Growth medium was completely removed, the cells were rinsed with PBS and fixed with 4% paraformaldehyde for 15 min. To suppress non-specific binding, animal serum was used and the cells were incubated with anti-TLR4 (Abcam; Cambridge, UK, dilution rate 1:200) and anti-RAGE (Santa-cruz biotechnology; Starr County, TX, USA, dilution rate 1:100) antibodies for 24 h at 4°C. After incubation, the cells were rinsed with PBS and incubated with Alexa 488–conjugated secondary antibody (dilution, 1:500) in the dark for 1 h and rinsed with PBS again. To stain the nuclei, the cells were incubated with DAPI (Sigma-Aldrich) for 1 min. Fluorescence was detected by confocal microscopy (LSM 710; Carl Zeiss, Oberkochen, Germany).

### 3.4. Immunoblotting

For protein extraction, TCMK-1 cells were lysed using an EzRIPA lysis kit (ATTO, Tokyo, Japan) and centrifuged at 13,000× *g* for 20 min at 4 °C. The clear supernatants were collected, and protein concentration was determined by using a bicinchoninic acid assay kit (Thermo Fisher Scientific). Proteins (15 µg/lane) were separated by sodium dodecylsulfate-polyacrylamide gel electrophoresis (SDS-PAGE) in a 10% to 12% gradient gels and transferred to polyvinylidene fluoride membranes in a Semi-Dry transfer system (ATTO) at 25 V for 10 min. The membranes were blocked with 5% skimmed milk for 1 h, washed twice with Tris-buffered saline containing 0.1% Tween 20 (TBST), and incubated with primary antibodies at 4 °C for 24 h. The membranes were then washed with TBST twice, incubated with appropriate secondary antibodies for 1 h at room temperature, and washed with TBST again. Proteins of interest were detected using ECL Western Blotting Substrate (Thermo Scientific; Waltham, MA, USA) on a LAS-4000 imager (GE Healthcare, Uppsala, Sweden). Protein expression levels were quantified using Image J software (NIH; Bethesda, MD, USA) and the antibodies used are listed in Appendix A.

### 3.5. Detection of Apoptotic Cells by TUNEL Staining

TUNEL staining was used to determine the effects of the 4 phlorotannins on apoptosis in both hypoxia treatment models. TCMK-1 cells were cultured in an 8-well chamber (3 × 10^5^ well) and treated with 4 phlorotannins before and after hypoxia for 24 h each., Cells were fixed with freshly prepared 4% paraformaldehyde for 5 min at room temperature, permeabilized with 0.1% Triton X-100 and 0.1% sodium citrate in PBS at 4 °C for 2 min and stained with the TUNEL assay kit (Roche; Indianapolis, IN, USA) following the manufacturer’s instructions. DAPI (Sigma-Aldrich) staining at room temperature for 5 min was used to visualize nuclei. Then, cover slips were mounted onto glass slides with Vectashield mounting medium (Vector Laboratories; San Francisco, CA, USA, H-1500) and examined under a confocal microscope (LSM 710). Apoptotic cells were analyzed with Zen 2009 software (Zeiss).

### 3.6. Enzyme-Linked Immunosorbent Assay (ELISA)

HMGB1 protein levels in cell lysates and culture supernatants of TCMK-1 cells exposed to hypoxia were measured by an indirect ELISA. A 96-well plate was coated with 0.6% sodium bicarbonate and 0.3% sodium carbonate in distilled water (pH 9.6) overnight at 4 °C. Each well was washed with 0.1% Triton X-100 in phosphate-buffered saline (TPBS), and remaining protein-binding sites were blocked with 5% skim milk at 4 °C overnight. The wells were washed with TPBS, and cell lysate and supernatant were added and incubated overnight at 4 °C. Unbound proteins were removed by washing with TPBS, and bound proteins were incubated with anti-HMGB1 antibody (Abcam, dilution rate 1:1000) for 24 h at 4 °C. After rinsing the plate with TPBS, the samples were incubated for 2 h at room temperature with horseradish peroxidase–conjugated anti-rabbit antibody (Vector, dilution rate 1:1000). After washing with TPBS, color was developed by incubating samples with 3,3′,5,5′-tetramethylbenzidine (Sigma-Aldrich) for 20 min. The reaction was stopped by adding 50 μL of 2 M H_2_SO_4_ to each well, and absorbance was measured at 450 nm using an ELISA plate reader (VERSA Max; Molecular Devices, San Jose, CA, USA).

### 3.7. Isolation of 4 Phlorotannins from E. cava

*E. cava* used this study were obtained from Aqua Green Technology Co., Ltd (Jeju, Korea). In briefly, *E. cava* were thoroughly washed with pure water and air-dried at room temperature for 48 h, the clean *E. cava* was ground and 50% ethyl alcohol was added followed by incubation at 85 °C for 12 h. The extracts of *E. cava* were filtered, concentrated, sterilized by heating at high temperatures above 85 °C for 40–60 min and then spray-dried. Phlorotannins were isolated following a published procedure [33]. In briefly, centrifugal partition chromatography (CPC) was performed using a two phase solvent system full of water/ethyl acetate/methyl alcohol/n-hexane (7:7:3:2, ratio v/v/v/v). The organic stationary phase was filled in the centrifugal partition chromatography column followed by pumping of the mobile phase into the column in descending mode at the same flow rate (2 mL per min) used for separation.

### 3.8. Statistical Analysis

The statistical analysis was conducted using SPSS version 22 software (IBM Co.; Armonk, NY, USA). In this study, statistical differences were compared among 6 groups (PBS, Hx/PBS, Hx/DK, Hx/PHB, Hx/PPB, Hx/PFFA) in pre-hypoxia and post-hypoxia treatment using the non-parametric Kruskal-Wallis test. For post-test, the difference between 2 groups was compared using the Mann-Whitney U test. All experiments were performed in triplicate, and results are presented as means ± standard deviation. * (asterisk) means comparing with PBS, ^$^ comparing with Hx/PBS and ^#^ comparing with Hx/PPB. All the experiments were repeated three times.

## 4. Conclusions

In conclusion, PPB showed a protective effect against H/R injury of TECs. The PPB reduced HMGB1 release, which initiates TLR4 and RAGE activation. The decrease in TRL4 and RAGE signaling by PPB inhibited NF-κB activation, which leads to TNF-α and IL-6 release. Those cytokines are well-known inducers of TEC apoptosis after H/R injury. PPB reduced TNF-α and IL-6 release, and reduced cell death and cell death signaling molecules such as pErk1/2, pSAPK/JNK, and pP38, which suggested that PPB has a potential to prevent kidney H/R injury by inhibiting the HMGB1-TLR4 or RAGE pathway. Among the 4 phlorotannins, PPB had the strongest protective effect from H/R injury, in both the pre- and post-hypoxia models. Our data suggested that PPB treatment could be used to protect against I/R injury during kidney surgery before the blood supply to the kidney is stopped and even after ischemia, before the blood supply to the kidney is restored, the latter treatment might be as effective as when treatment with PPB is performed before the onset of ischemia.

## Figures and Tables

**Figure 1 marinedrugs-17-00602-f001:**
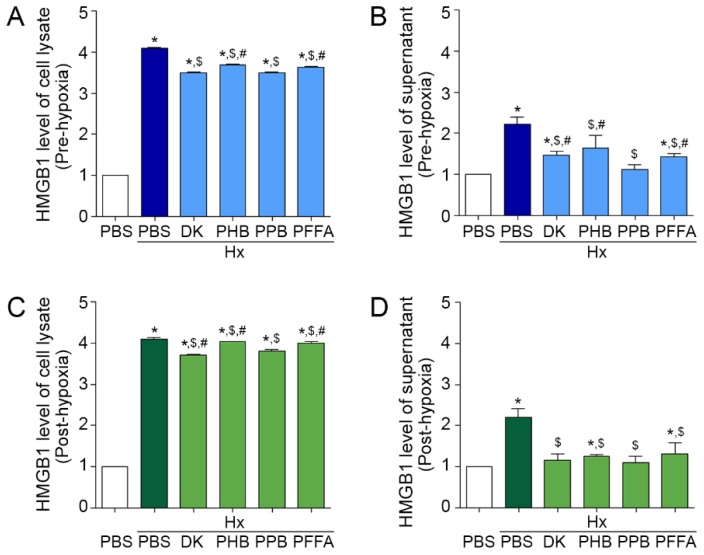
Inhibitory effects of phlorotannins from *E. cava* extract on HMGB1 synthesis and secretion in pre-hypoxia and post-hypoxia treatment (**A**,**B**) To examine the preventive effects of 4 phlorotannins from *E. cava* extract (DK, PHB, PPB, PFFA), they were added to mouse kidney tubular cells (TCMK-1) before hypoxia (pre-hypoxia treatment). (**C**,**D**) To examine the therapeutic effects of the 4 phlorotannins, they were added to TCMK-1 after hypoxia (post-hypoxia treatment). In each treatment model, (**A**,**C**) HMGB1 synthesis in cell lysate and (**B**,**D**) secretion levels in cell culture medium were measured by ELISA. All levels are normalized to those in cells treated with PBS under normoxic control conditions. Significance represented as: * *p* < 0.05 versus PBS, ^$^
*p* < 0.05 versus Hx/PBS, ^#^
*p* < 0.05 versus Hx/PPB. DK, dieckol, PHB, 2,7-phloroglucinol-6,6-bieckol, PPB, pyrogallol-phloroglucinol-6,6-bieckol, PFFA, phlorofucofuroeckol A, HMGB1, high mobility group box 1.

**Figure 2 marinedrugs-17-00602-f002:**
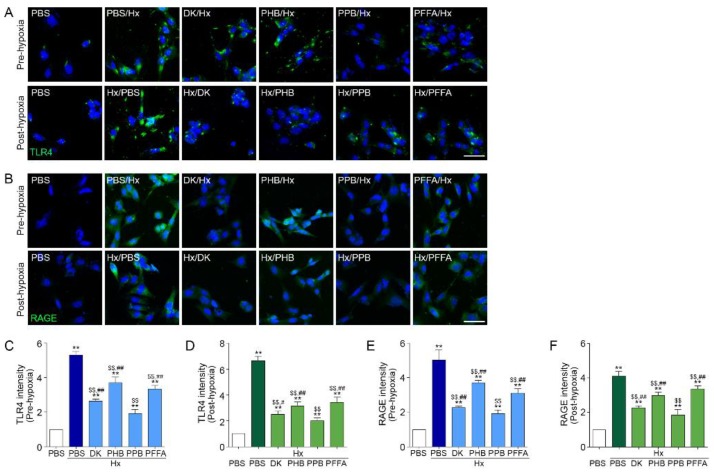
Inhibitory effects of phlorotannins from *E. cava* extract on TLR4 and RAGE expression in pre-hypoxia and post-hypoxia treatment using Mouse kidney tubular cells (TCMK-1) were treated with 4 phlorotannins (DK, PFFA, PPB, PHB) before hypoxia (pre-hypoxia model) or after hypoxia (post-hypoxia model). (**A**,**B**) Microscopic fluorescence images showing (**A**) TLR4 and (**B**) RAGE protein expression (green) in the pre-hypoxia treatment (upper rows) and post-hypoxia treatment (bottom rows). Nuclei were stained with DAPI (blue). Scale bar = 50 µm. (**C**–**F**) Quantification of (**C**,**D**) TLR4 and (**E**,**F**) RAGE expression using representative images from (**A**) and (**B**) in the pre-hypoxia treatment model and post-hypoxia treatment. Significance represented as: ** *p* < 0.01 versus PBS, ^$$^
*p* < 0.01 versus Hx/PBS, ^#^
*p* < 0.05, ^##^
*p* < 0.01 versus Hx/PPB. DK, dieckol, PHB, 2,7-phloroglucinol-6,6-bieckol, PFFA, phlorofucofuroeckol A, PPB, pyrogallol-phloroglucinol-6,6-bieckol, TLR4, Toll-like receptor 4, RAGE, receptor for advanced glycation end-products.

**Figure 3 marinedrugs-17-00602-f003:**
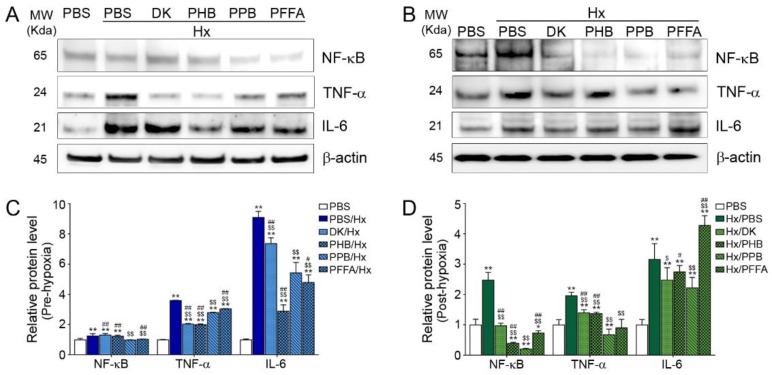
Inhibitory effects of phlorotannins from *E. cava* extract on NF-kB expression and pro-inflammatory cytokine expression in pre-hypoxia and post-hypoxia treatment Mouse kidney tubular cells (TCMK-1) were treated with 4 phlorotannins (DK, PFFA, PPB, PHB) before hypoxia (pre-hypoxia treatment) or after hypoxia (post-hypoxia treatment). NF-κB and the pro-inflammatory cytokines TNF-α and IL-6 were detected by immunoblotting in (**A**) pre-hypoxia and (**B**) post-hypoxia treatment and immunoblotting results are quantified and normalized to those in cells treated with PBS in (**C**) pre-hypoxia and (**D**) post-hypoxia treatment. Significance represented as: * *p* < 0.05, ** *p* < 0.01 versus PBS, ^$^
*p* < 0.05, ^$$^
*p* < 0.01 versus Hx/PBS, ^#^
*p* < 0.05, ^##^
*p* < 0.01 versus Hx/PPB. DK, dieckol, Hx, Hypoxia, PHB, 2,7-phloroglucinol-6,6-bieckol, PFFA, phlorofucofuroeckol A, PPB, pyrogallol-phloroglucinol-6,6-bieckol, NF-kB, nuclear factor kappa light chain enhancer of activated B cells, TNF-α, tumor necrosis factor-alpha, IL-6, Interleukin-6.

**Figure 4 marinedrugs-17-00602-f004:**
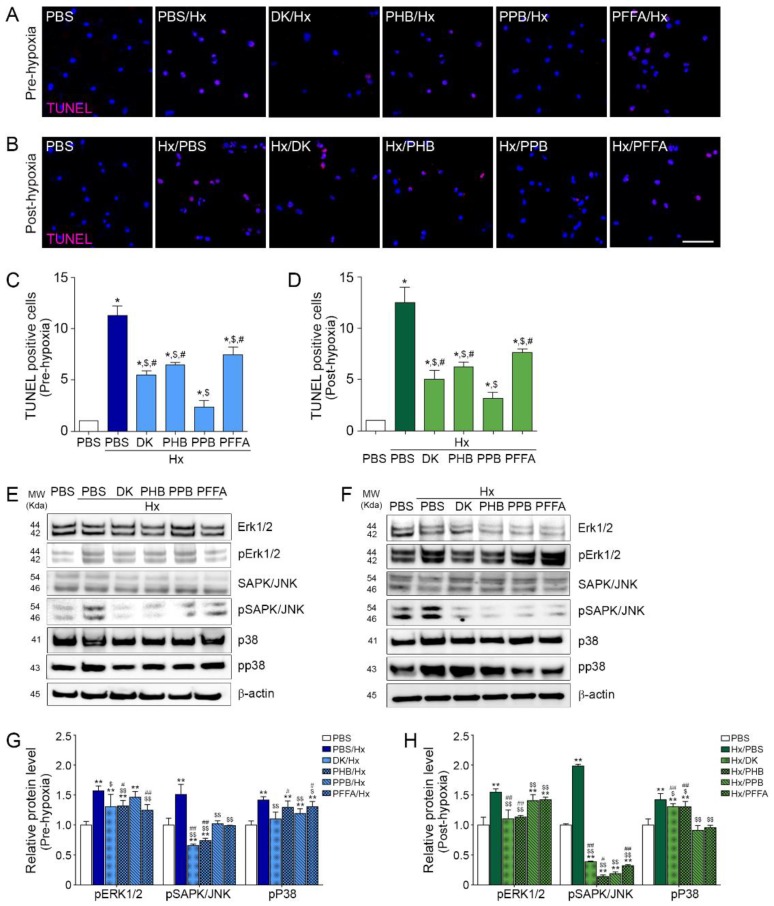
Inhibitory effects of phlorotannins from *E. cava* extract on apoptosis and apoptotic cell death–related MAPK pathway molecules in pre-hypoxia and post-hypoxia treatment Mouse kidney tubular cells (TCMK-1) were treated with the 4 phlorotannins (DK, PHB, PPB, PFFA) before hypoxia (pre-hypoxia treatment) or after hypoxia (post-hypoxia treatment). (**A**,**B**) Microscopic fluorescence images showing TUNEL-positive apoptotic cells (pink dot) and nuclei (blue, DAPI) in (**A**) the pre-hypoxia treatment and **(B)** post-hypoxia treatment. Scale bar = 50 µm. Quantification of TUNEL-positive apoptotic cells using representative images is shown in (**C**) the pre-hypoxia treatment and in (**D**) the post-hypoxia treatment. Apoptosis-related MAPK pathway molecules were detected by immunoblotting in (**E**) pre-hypoxia treatment and (**F**) post-hypoxia treatment. Apoptosis-related phosphorylated MAPKs expression are quantified using non-phosphorylated protein expression in (**G**) pre-hypoxia and (**H**) post-hypoxia treatment model. All levels are normalized to those in cells treated with PBS. Significance represented as: * *p* < 0.05, ** *p* < 0.01 versus PBS, ^$^
*p* < 0.05, ^$$^
*p* < 0.01 versus Hx/PBS, ^#^
*p* < 0.05, ^##^
*p* < 0.01 versus Hx/PPB. DK, dieckol, Hx, Hypoxia, PHB, 2,7-phloroglucinol-6,6-bieckol, PFFA, phlorofucofuroeckol A, PPB, pyrogallol-phloroglucinol-6,6-bieckol, TUNEL, terminal deoxynucleotidyl transferase dUTP nick end labeling.

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
