# Peer review of "Pyrogallol-Phloroglucinol-6,6-Bieckol from Ecklonia cava Attenuates Tubular Epithelial Cell (TCMK-1) Death in Hypoxia/Reoxygenation Injury"

_marinedrugs, 2019, doi:10.3390/md17110602_

Round 1

Reviewer 1 Report

the work is difficult to follow due to the use of too many acronyms.

The authors in the title argue that this in vitro model can be comparable with  in vivo renal ischemia-reperfusion damage

I do not believe that this model can represent the situation in vivo. Renal tubular cells are sensitive to factors inducing inflammation and in particular aldosterone, erithrocytes platelets other circulating factors. It is known that the NFKb factor and other cytokines  are induced especially in inflammatory cells (different T lymphocytes ) T lymphocytes are not present in this in vivo model. I suggest rewriting the work considering how an ischemia-inducing medium or not can be influenced by the co-incubation with the extracts of e. cava

he authors should explain how many experiments have been done for each incubation (duplicate? single experiment?)

in conclusion the study is of interest but the rationale , introduction and discussion should be rewritten

Author Response

Response to Reviewer 1 Comments

Point 1: the work is difficult to follow due to the use of too many acronyms.

Response 1: We appreciated and agree with this comment. When we looked thoroughly all manuscript, we found that many acronyms and it could confuse the reader. So, we listed acronyms in abbreviation section of manuscript (Abbreviation section, Line 329).

Manuscript: Abbreviations, Line 329 (previous)

DK, dieckol; PHB, 2,7-phloroglucinol-6,6-bieckol; PPB, pyrogallol-phloroglucinol-6,6-bieckol; PFFA, phlorofucofuroeckol A

Manuscript: Abbreviations, Line 329 (New)

DK, dieckol; Hx, hypoxia; H/R, hypoxia/ reoxygenation; I/R, ischemia/reperfusion; PFFA, phlorofucofuroeckol A; PHB, 2,7-phloroglucinol-6,6-bieckol; PPB, pyrogallol-phloroglucinol-6,6-bieckol; RAGE, receptor for advanced glycation end-products; TEC, kidney tubular epithelial cells; TLR4, Toll-like receptor 4; TUNEL, terminal deoxynucleotidyl transferase dUTP nick end labeling

Point 2: The authors in the title argue that this in vitro model can be comparable with  in vivo renal ischemia-reperfusion damage. I do not believe that this model can represent the situation in vivo. Renal tubular cells are sensitive to factors inducing inflammation and in particular aldosterone, erithrocytes platelets other circulating factors. It is known that the NFKb factor and other cytokines  are induced especially in inflammatory cells (different T lymphocytes ) T lymphocytes are not present in this in vivo model. I suggest rewriting the work considering how an ischemia-inducing medium or not can be influenced by the co-incubation with the extracts of e. cava

Response 2: We appreciated this comment. As this comment, it is not easy to accurately implement ischemia/reperfusion (I/R) in in vitro cell experiments. However, because the researchers used hypoxia /reoxygenation (H/R) to implement I/R in in vitro cell experiments [Ref; Cell Death Dis. 2018 Mar 1;9(3):338 & J Pharmacol Sci. 2018 Jun;137(2):170-176.], and we also used the same method, we thought that our model was an I/R model. However, in order to make the reader's understanding more accurate, we revised all of the I/R to H/R in manuscript including title and added details to the Materials and methods section.

Manuscript: title (previous)

Pyrogallol-phloroglucinol-6,6-bieckol from Ecklonia cava attenuates kidney tubular cell death in in vitro renal ischemia/reperfusion models

Manuscript: title (new)

Pyrogallol-phloroglucinol-6,6-bieckol from Ecklonia cava attenuates tubular epithelial cell (TCMK-1) death in hypoxia/reoxygenation injury

Manuscript: Abstract, Line 25 (previous)

The ischemia/reperfusion (I/R) injury causes serious complications after the blood supply to the kidney is stopped during surgery. The main mechanism of I/R injury is the release of high-mobility group protein B1 (HMGB1) from injured tubular epithelial cells (TEC), which triggers TLR4 or RAGE signaling, leading to cell death. We evaluated whether the extracts of Ecklonia cava would attenuate TEC death induced by I/R injury. We also evaluated which phlorotannin—dieckol (DK), phlorofucofuroeckol A (PFFA), pyrogallol phloroglucinol-6,6-bieckol (PPB), or 2,7-phloroglucinol-6,6-bieckol (PHB)—would have the most potent effect in the context of I/R injury. We used a pre-ischemia model, in which the E. cava extracts were added before the onset of ischemia, and a post-ischemia model, in which the extracts were added before start of reperfusion. PPB most effectively reduced HMGB1 release and the expression of TLR4 and RAGE induced by I/R injury in both pre- and post-ischemia models. PPB also most effectively inhibited expression of NF-kB and release of the inflammatory cytokines TNF-α and IL-6 in both models. PPB most effectively inhibited cell death and expression of cell death signaling molecules such as Erk/pErk, JNK/pJNK, and p38/pp38. These results suggest that PPB blocks the HGMB1–TLR4/RAGE signaling pathway and decreases TEC death induced by I/R, and that PPB can be a novel target for renal I/R injury therapy

Manuscript: Abstract, Line 25 (new)

The hypoxia/reoxygenation (H/R) injury causes serious complications after the blood supply to the kidney is stopped during surgery. The main mechanism of I/R injury is the release of high-mobility group protein B1 (HMGB1) from injured tubular epithelial cells (TEC, TCMK-1 cell), which triggers TLR4 or RAGE signaling, leading to cell death. We evaluated whether the extracts of Ecklonia cava (E. cava) would attenuate TEC death induced by H/R injury. We also evaluated which phlorotannin—dieckol (DK), phlorofucofuroeckol A (PFFA), pyrogallol phloroglucinol-6,6-bieckol (PPB), or 2,7-phloroglucinol-6,6-bieckol (PHB)—would have the most potent effect in the context of H/R injury. We used for pre-hypoxia treatment, in which the phlorotannins from E. cava extracts were added before the onset of hypoxia, and a post- hypoxia treatment, in which the phlorotannins were added before start of reperfusion. PPB most effectively reduced HMGB1 release and the expression of TLR4 and RAGE induced by H/R injury in both pre- and post- hypoxia treatment. PPB also most effectively inhibited expression of NF-kB and release of the inflammatory cytokines TNF-α and IL-6 in both models. PPB most effectively inhibited cell death and expression of cell death signaling molecules such as Erk/pErk, JNK/pJNK, and p38/pp38. These results suggest that PPB blocks the HGMB1–TLR4/RAGE signaling pathway and decreases TEC death induced by H/R, and that PPB can be a novel target for renal H/R injury therapy.

Manuscript: Introduction, Line 64 (previous)

Here, we evaluated whether extracts of E. cava would attenuate TEC death induced by I/R injury. We used two models: a pre-ischemia model, in which the E. cava extract was added before the onset of ischemia, and a post-ischemia model, in which the E. cava extract was added before the start of reperfusion. In addition, we evaluated which phlorotannin—dieckol (DK), phlorofucofuroeckol A (PFFA), pyrogallol phloroglucinol-6,6-bieckol (PPB), or 2,7-phloroglucinol-6,6-bieckol (PHB) —would have the most potent effect in the context of I/R injury.

Manuscript: Introduction, Line 64 (new)

Here, we evaluated whether phlorotannins from E. cava extract would attenuate TEC death induced by I/R injury. Commonly, renal hypoxia/reoxygenation (H/R) was established to simulate renal I/R injury in vitro [16, 17] and we also used H/R model. We used two treatment models: a pre-hypoxia model, in which the phlorotannins was added before the onset of hypoxia, and a post- hypoxia model, in which the phlorotannins was added before the start of reoxygenation. In addition, we evaluated which phlorotannin—dieckol (DK), phlorofucofuroeckol A (PFFA), pyrogallol phloroglucinol-6,6-bieckol (PPB), or 2,7-phloroglucinol-6,6-bieckol (PHB) —would have the most potent effect in the context of H/R injury.

Manuscript: Results and Discussion, Line 73 (previous)

2.1. Attenuation of HMGB1 release from TECs after I/R injury by E. cava extracts

In this study, we used mouse kidney tubular cells (TCMK-1) as TECs. In the pre-ischemia model, the HMGB1 level was increased by I/R injury in both TEC lysate and supernatant (Fig. 1A, B), suggesting that TECs injured by I/R increased the synthesis and release of HMGB1. HMGB1 levels in both the TEC lysate and supernatant were decreased by individual phlorotannins added before TECs were exposed to ischemia; among individual phlorotannins, PPB and DK showed the strongest attenuation effects.

In the post-ischemia model, the HMGB1 level increased by I/R injury was also decreased by adding E. cava extracts before reperfusion in both TEC lysate and supernatant. Among the 4 phlorotannins, the effect of DK and PPB was the  significant (Fig. 1C, D). HMGB1 is passively released in response to inflammatory stress or necrosis [16]. Our results showed that the HMGB1 release from TECs was increased after I/R injury and that this increase was attenuated by PPB most significantly among the 4 E. cava phlorotannins.

Manuscript: Results and Discussion, Line 73 (new)

2.1. Attenuation of HMGB1 release from TECs after H/R injury by the phlorotannins from E. cava extracts

In this study, we used mouse kidney tubular cells (TCMK-1) as TECs. In the pre-hypoxia model, the HMGB1 level was increased by H/R injury in both TEC lysate and supernatant (Fig. 1A, B), suggesting that TECs injured by H/R increased the synthesis and release of HMGB1. HMGB1 levels in both the TEC lysate and supernatant were decreased by individual phlorotannins added before TECs were exposed to hypoxia; among individual phlorotannins, PPB and DK showed the strongest attenuation effects.

In the post-hypoxia treatment model, the HMGB1 level increased by H/R injury was also decreased by adding E. cava extracts before reperfusion in both TEC lysate and supernatant. Among the 4 phlorotannins, the effect of DK and PPB was the most significant (Fig. 1C, D). HMGB1 is passively released in response to inflammatory stress or necrosis [18]. Our results showed that the HMGB1 release from TECs was increased after H/R injury and that this increase was attenuated by PPB most significantly among the 4 phlorotannins from E. cava.

Manuscript: Results and Discussion, Line 104 (previous)

2.2. TLR4 and RAGE expression induced by I/R injury is attenuated most efficiently by PPB

TLR4 and RAGE expression in TECs induced by I/R injury was attenuated by the 4 phlorotannins in the pre-ischemia model; among them, PPB had the strongest effect on TLR4 and RAGE expression (Fig. 2A, B, C, E). The patterns were the same in the post-ischemia model (Fig. 2A, B, D, F).

TLR4 [17] and RAGE [18] are the primary cell membrane receptors that bind HMGB1. In I/R injury, this binding leads to pro-inflammatory signaling pathway activation by TLR and RAGE.

Manuscript: Results and Discussion, Line 104 (new)

2.2. TLR4 and RAGE expression induced by H/R injury is attenuated most efficiently by PPB

TLR4 and RAGE expression in TECs induced by H/R injury was attenuated by the 4 phlorotannins in the pre-hypoxia treatment model; among them, PPB had the strongest effect on TLR4 and RAGE expression (Fig. 2A-C, E). The patterns were the same in the post-hypoxia treatment (Fig. 2A, B, D, F).

TLR4 [19] and RAGE [20] are the primary cell membrane receptors that bind HMGB1. In H/R injury, this binding leads to pro-inflammatory signaling pathway activation by TLR and RAGE.

Manuscript: Results and Discussion, Line 126 (previous)

2.3. Expression of NF-kB and pro-inflammatory cytokines after I/R injury is attenuated efficiently by PPB

The level of NF-κB was increased in both the pre-ischemia and post-ischemia models and was decreased by the 4 phlorotannins, with PPB showing the strong effect (Fig. 3). The levels of TNF-α and IL-6 were also increased by I/R injury and were attenuated by adding the 4 phlorotannins, among which PPB showed the strong effect.

During I/R injury, TLR4 initiates the inflammatory response by increasing the production of NF-κB-dependent cytokines such as TNF-α [5,6]. In addition, IL-6 is released upon TLR4 activation during I/R injury and amplifies inflammation [19,20]. The RAGE pathway also induces NF-κB activation and leads to an increase in TNF-α and IL-6 levels [21]. TNF-α and IL-6 induce tubular cell death during renal I/R injury [22].

Manuscript: Results and Discussion, Line 126 (new)

2.3. Expression of NF-kB and pro-inflammatory cytokines after H/R injury is attenuated efficiently by PPB

The level of NF-κB was increased in both the pre-hypoxia and post-hypoxia treatment models and was decreased by the 4 phlorotannins, with PPB showing the strong effect (Fig. 3). The levels of TNF-α and IL-6 were also increased by H/R injury and were attenuated by adding the 4 phlorotannins, among which PPB showed the strong effect.

During H/R injury, TLR4 initiates the inflammatory response by increasing the production of NF-κB-dependent cytokines such as TNF-α [5,6]. In addition, IL-6 is released upon TLR4 activation during H/R injury and amplifies inflammation [21,22]. The RAGE pathway also induces NF-κB activation and leads to an increase in TNF-α and IL-6 levels [23]. TNF-α and IL-6 induce tubular cell death during renal H/R injury [24].

Manuscript: Results and Discussion, Line 149 (previous)

2.4. TEC death induced by I/R injury is attenuated efficiently by PPB

TEC apoptosis induced by I/R injury was attenuated by adding E. cava extracts in both the pre-ischemia and post-ischemia models.

Manuscript: Results and Discussion, Line 149 (new)

2.4. TEC apoptosis induced by H/R injury is attenuated efficiently by PPB

TEC apoptosis induced by H/R injury was attenuated by adding the phlorotannin from E. cava extracts in both the pre-hypoxia and post-hypoxia treatments.

Manuscript: Results and Discussion, Line 166 (previous)

2.5. PPB attenuated cell death signals which induced by I/R injury

The extracellular signal-regulated kinases-1 and -2 (Erk1/2), the c-Jun N-terminal kinases (SAPK/JNK), and p38 mitogen-activated protein kinases (MAPKs) are involved in cell death induced by I/R injury [23]. ERK signaling is involved in injury and apoptosis of kidney cells during I/R injury [24-26]. Both p38 and JNK pathways induce tubular cell death and their inhibition reduce apoptosis and inflammation induced by I/R injury [27,28].

In our study, these cell death signals were increased by I/R injury in TEC, and were attenuated most significantly by PPB in both the pre-ischemia and post-ischemia models.

Manuscript: Results and Discussion, Line 166 (new)

2.5. PPB attenuated cell death signals induced by H/R injury

The extracellular signal-regulated kinases-1 and -2 (Erk1/2), the c-Jun N-terminal kinases (SAPK/JNK), and p38 mitogen-activated protein kinases (MAPKs) are involved in cell death induced by H/R injury [25]. ERK signaling is involved in injury and apoptosis of kidney cells during H/R injury [26-28]. Both p38 and JNK pathways induce tubular cell death and their inhibition reduce apoptosis and inflammation induced by H/R injury [29,30].

In our study, these cell death signals were increased by H/R injury in TEC, and were attenuated most significantly by PPB in both the pre-hypoxia and post-hypoxia treatment models.

Manuscript: Results and Discussion, Line 176 (previous)

3.2. Kidney ischemia reperfusion cell models

To examine the inhibitory effects of 4 phlorotannins DK, PHB, PFFA and PPB from E. cava extract, we designed two types of ischemia/reperfusion cell models using TCMK-1 cells; the experimental scheme is shown in Figure S1.

3.2.1. Pre-ischemia model

Four prepared phlorotannins (2.5 μg/ml) were added to growth medium, which was used to treat TCMK-1 cells for 6 h before ischemia. After treatment, the medium was completely removed, and the cells were washed thoroughly with PBS. Ischemia medium contained 0.5% FBS and 1% penicillin/streptomycin in DMEM. The TCMK-1 cells were cultured in ischemia medium with the 4 phlorotannins and exposed to a mixture of 1% O2, 5% CO2, 94% N2 for 14 h. After incubation, the ischemia medium was removed, and the cells were washed with PBS. The cells were cultured in growth medium containing 4 phlorotannins in a 5% CO2 incubator for 9 h, and then used for experiments.

3.2.2. Post-ischemia model

TCMK-1 cells were cultured in growth medium for 6 h before ischemia, growth medium was completely removed, and the cells were washed with PBS. Ischemia treatment was performed as in 3.2.1. After the treatment, the ischemia medium was removed, and the cells were washed with PBS. The cells were cultured in growth medium containing 4 phlorotannins in a 5% CO2 incubator for 9 h, and then used for experiments.

Manuscript: Materials and methods, Line 176 (new)

3.2. Hypoxia/reoxygenation injury cell models

To examine the inhibitory effects of 4 phlorotannins DK, PHB, PFFA and PPB from E. cava extract, we designed two types of H/R cell models using TCMK-1 cells; the experimental method and image scheme is shown in Figure S1.

3.2.1. Pre-hypoxia treatment model

Four prepared phlorotannins (2.5 μg/ml) were added to growth medium, which was used to treat TCMK-1 cells for 6 h before hypoxia. After treatment, the medium was completely removed, and the cells were washed thoroughly with PBS. Hypoxia medium contained 0.5% FBS and 1% penicillin/streptomycin in DMEM. The TCMK-1 cells were cultured in hypoxia medium with the 4 phlorotannins and exposed to a mixture of 1% O2, 5% CO2, 94% N2 for 14 h. After incubation, the hypoxia medium was removed, and the cells were washed with PBS. The cells were cultured with fresh oxygenated medium containing FBS, 1% penicillin/streptomycin, 4 phlorotannins in a 5% CO2 incubator for 9 h, and then used for experiments.

3.2.2. Post-hypoxia treatment model

TCMK-1 cells were cultured in growth medium for 6 h before ischemia, growth medium was completely removed, and the cells were washed with PBS. Hypoxia treatment was performed as in 3.2.1 of material and methods section. After the hypoxia treatment, the hypoxia medium was removed, and the cells were washed with PBS. The cells were cultured the fresh oxygenated medium containing 4 phlorotannins in a 5% CO2 incubator for 9 h, and then used for experiments.

Manuscript: Material and methods, Line 191 (previous)

3.2. Kidney ischemia reperfusion cell models

To examine the inhibitory effects of 4 phlorotannins DK, PHB, PFFA and PPB from E. cava extract, we designed two types of ischemia/reperfusion cell models using TCMK-1 cells; the experimental scheme is shown in Figure S1.

3.2.1. Pre-ischemia model

Four prepared phlorotannins (2.5 μg/ml) were added to growth medium, which was used to treat TCMK-1 cells for 6 h before ischemia. After treatment, the medium was completely removed, and the cells were washed thoroughly with PBS. Ischemia medium contained 0.5% FBS and 1% penicillin/streptomycin in DMEM. The TCMK-1 cells were cultured in ischemia medium with the 4 phlorotannins and exposed to a mixture of 1% O2, 5% CO2, 94% N2 for 14 h. After incubation, the ischemia medium was removed, and the cells were washed with PBS. The cells were cultured in growth medium containing 4 phlorotannins in a 5% CO2 incubator for 9 h, and then used for experiments.

3.2.2. Post-ischemia model

TCMK-1 cells were cultured in growth medium for 6 h before ischemia, growth medium was completely removed, and the cells were washed with PBS. Ischemia treatment was performed as in 3.2.1. After the treatment, the ischemia medium was removed, and the cells were washed with PBS. The cells were cultured in growth medium containing 4 phlorotannins in a 5% CO2 incubator for 9 h, and then used for experiments.

Manuscript: Material and methods, Line 191 (new)

3.2. Hypoxia/reoxygenation (H/R) injury cell models

To examine the inhibitory effects of 4 phlorotannins DK, PHB, PFFA and PPB from E. cava extract, we designed two types of H/R cell models using TCMK-1 cells; the experimental method and image scheme is shown in Figure S1.

3.2.1. Pre-hypoxia treatment model

Four prepared phlorotannins (2.5 μg/ml) were added to growth medium, which was used to treat TCMK-1 cells for 6 h before hypoxia. After treatment, the medium was completely removed, and the cells were washed thoroughly with PBS. Hypoxia medium contained 0.5% FBS and 1% penicillin/streptomycin in DMEM. The TCMK-1 cells were cultured in hypoxia medium with the 4 phlorotannins and exposed to gas mixture of 1% O2, 5% CO2, 94% N2 for 14 h within hypoxia incubator chamber (Stemcell technologies; MA, USA). After flushing the hypoxia chamber with the desired gas mixture, the chamber seals tightly with clamp. After incubation, the hypoxia chamber was dismantled and the hypoxia medium was removed, and the cells were washed with PBS. The cells were cultured with fresh oxygenated medium containing FBS, 1% penicillin/streptomycin, 4 phlorotannins in a mixture of 20% O2, 5% CO2, 75% N2 for 9 h, and then used for experiments. The experimental method and image scheme is shown in Figure S1.

3.2.2. Post-hypoxia treatment model

TCMK-1 cells were cultured in growth medium for 6 h before hypoxia, growth medium was completely removed, and the cells were washed with PBS. Hypoxia exposure was performed as in 3.2.1. Pre-hypoxia treatment model of material and methods section. After the hypoxia treatment, the hypoxia chamber was dismantled and the hypoxia medium was removed, and the cells were washed with PBS. The cells were cultured with fresh oxygenated medium containing FBS, 1% penicillin/streptomycin, 4 phlorotannins in a mixture of 20% O2, 5% CO2, 75% N2 for 9 h, and then used for experiments. The experimental method and image scheme is shown in Figure S1.

Point 3: he authors should explain how many experiments have been done for each incubation (duplicate? single experiment?)

Response 3: We appreciated this comment. As this comment, all the experiments were repeated three times and all of them produced the pre-hypoxia and post-hypoxia treatment in the same way. This content was added to the materials and methods section (Materials and methods section, Line 261).

Manuscript: Materials and methods, Line 261 (previous)

3.7. Statistical analysis

The Kruskal–Wallis test and the Mann–Whitney U post-hoc test were used to determine the significance of differences among the normoxia control, ischemia/reperfusion, and ischemia/reperfusion with phlorotannins groups. Results are presented as mean ± SD, and statistical significance was accepted for p values <0.05. The analysis was performed using SPSS version 22 (IBM Corporation, NY, USA). Same letters represent no significant difference (p ≥ 0.05).

Manuscript: Materials and methods, Line 261 (new)

3.8. Statistical analysis

The statistical analysis was conducted using SPSS version 22 software (IBM Co.; NY, USA). In this study, statistical differences were compared among 6 groups (PBS, Hx/PBS, Hx/DK, Hx/PHB, Hx/PPB, Hx/PFFA) in pre-hypoxia and post-hypoxia treatment using the non-parametric Kruskal-Wallis test. For post-test, difference between 2 groups was compared using the Mann-Whitney U test. All experiments were performed in triplicate, and results are presented as means ± standard deviation. * (asterisk) means comparing with PBS, $ comparing with Hx/PBS and # comparing  with Hx/PPB. All the experiments were repeated three times.

Point 4: in conclusion the study is of interest but the rationale, introduction and discussion should be rewritten

Response 4: We appreciated this comment. As this comment, rationale, introduction, and discussion are rewritten in manuscript.

Manuscript: Introduction section, Line 38 (previous)

1. Introduction

Ischemia/reperfusion (I/R) injury, which occurs when blood supply to tissues or organs is restored after ischemia, leads to more tissue damage than ischemia itself by enhancing the inflammatory reaction in the reperfused tissue [1].

Renal I/R injury is a major pathophysiology of acute kidney injury (AKI), and can induce AKI after kidney transplantation, partial nephrectomy, renal artery angioplasty, aortic aneurysm surgery, and elective urological surgery when blood supply to the kidney is stopped or decreased during surgery [2]. In addition to AKI, I/R injury leads to the loss of tubular epithelial cell (TEC) function, resulting in delayed graft function and acute or chronic rejection of the transplanted kidney [3]. To decrease the incidence of AKI after surgeries accompanied by I/R injury, treatments to decrease I/R injury should be developed.

High-mobility group protein B1 (HMGB1) is released from injured reunal cells and activates Toll-like receptors (TLR), which trigger production of pro-inflammatory cytokines such as tumor necrosis factor-α (TNF-α) and transcription of nuclear factor kappa B (NF-κB) [4-6], finally leading to tissue injury after I/R injury in the kidney [7,8]. In I/R injury, TECs play a dual role as both injury initiators (by releasing HMGB1) and targets [9]. Receptor for advanced glycation end products (RAGE) also initiates pro-inflammatory signaling in I/R injury by binding HMGB1 in the liver and heart [10,11].

Ecklonia cava is a brown alga that contains phlorotannins, polyphenolic compounds that have multiple biological activities including anti-inflammatory [12,13] and antioxidant activities [14]. One study has shown that polyphenol extract from E. cava attenuated renal inflammation induced by high-fat diet by decreasing pro-inflammatory signaling via TNF-α and NF-κB [15]. However, the effect of E. cava on I/R injury has not been studied.

Here, we evaluated whether extracts of E. cava would attenuate TEC death induced by I/R injury. We used two models: a pre-ischemia model, in which the E. cava extract was added before the onset of ischemia, and a post-ischemia model, in which the E. cava extract was added before the start of reperfusion. In addition, we evaluated which phlorotannin—dieckol (DK), phlorofucofuroeckol A (PFFA), pyrogallol phloroglucinol-6,6-bieckol (PPB), or 2,7-phloroglucinol-6,6-bieckol (PHB) —would have the most potent effect in the context of I/R injury.

Manuscript: Introduction section, Line 38 (new)

1. Introduction

Ischemia/reperfusion (I/R) injury, which occurs when blood supply to tissues or organs is restored after ischemia, leads to more tissue damage than ischemia itself by enhancing the inflammatory reaction in the reperfused tissue [1].

Renal I/R injury is a major pathophysiology of acute kidney injury (AKI), and can induce AKI after kidney transplantation, partial nephrectomy, renal artery angioplasty, aortic aneurysm surgery, and elective urological surgery when blood supply to the kidney is stopped or decreased during surgery [2]. In addition to AKI, I/R injury leads to the loss of tubular epithelial cell (TEC) function, resulting in delayed graft function and acute or chronic rejection of the transplanted kidney [3]. To decrease the incidence of AKI after surgeries accompanied by I/R injury, treatments to decrease I/R injury should be developed.

High-mobility group protein B1 (HMGB1) is released from injured renal cells and activates Toll-like receptors (TLR), which trigger production of pro-inflammatory cytokines such as tumor necrosis factor-α (TNF-α) and transcription of nuclear factor kappa B (NF-κB) [4-6], finally leading to tissue injury after I/R injury in the kidney [7,8]. In I/R injury, TECs play a dual role as both injury initiators (by releasing HMGB1) and targets [9]. Receptor for advanced glycation end products (RAGE) also initiates pro-inflammatory signaling in I/R injury by binding HMGB1 in the liver and heart [10,11].

Ecklonia cava is a brown alga that contains phlorotannins, polyphenolic compounds that have multiple biological activities including anti-inflammatory [12,13] and antioxidant activities [14]. One study has shown that polyphenol extract from E. cava attenuated renal inflammation induced by high-fat diet by decreasing pro-inflammatory signaling via TNF-α and NF-κB [15]. However, the effect of E. cava on I/R injury has not been studied.

Here, we evaluated whether phlorotannins from E. cava extract would attenuate TEC death induced by I/R injury. Commonly, renal hypoxia/reoxygenation (H/R) was established to simulate renal I/R injury in vitro [16, 17] and we also used H/R model. We used two treatment models: a pre-hypoxia model, in which the phlorotannins was added before the onset of hypoxia, and a post- hypoxia model, in which the phlorotannins was added before the start of reoxygenation. In addition, we evaluated which phlorotannin—dieckol (DK), phlorofucofuroeckol A (PFFA), pyrogallol phloroglucinol-6,6-bieckol (PPB), or 2,7-phloroglucinol-6,6-bieckol (PHB) would have the most potent effect in the context of H/R injury.

Manuscript: Results and discussion section, Line 73 (previous)

2. Results and Discussion

2.1. Attenuation of HMGB1 release from TECs after I/R injury by E. cava extracts

In this study, we used mouse kidney tubular cells (TCMK-1) as TECs. In the pre-ischemia model, the HMGB1 level was increased by I/R injury in both TEC lysate and supernatant (Fig. 1A, B), suggesting that TECs injured by I/R increased the synthesis and release of HMGB1. HMGB1 levels in both the TEC lysate and supernatant were decreased by individual phlorotannins added before TECs were exposed to ischemia; among individual phlorotannins, PPB and DK showed the strongest attenuation effects.

In the post-ischemia model, the HMGB1 level increased by I/R injury was also decreased by adding E. cava extracts before reperfusion in both TEC lysate and supernatant. Among the 4 phlorotannins, the effect of DK and PPB was the  significant (Fig. 1C, D).

HMGB1 is passively released in response to inflammatory stress or necrosis [16]. Our results showed that the HMGB1 release from TECs was increased after I/R injury and that this increase was attenuated by PPB most significantly among the 4 E. cava phlorotannins.

2.2. TLR4 and RAGE expression induced by I/R injury is attenuated most efficiently by PPB

TLR4 and RAGE expression in TECs induced by I/R injury was attenuated by the 4 phlorotannins in the pre-ischemia model; among them, PPB had the strongest effect on TLR4 and RAGE expression (Fig. 2A, B, C, E). The patterns were the same in the post-ischemia model (Fig. 2A, B, D, F).

TLR4 [17] and RAGE [18] are the primary cell membrane receptors that bind HMGB1. In I/R injury, this binding leads to pro-inflammatory signaling pathway activation by TLR and RAGE.

2.3. Expression of NF-kB and pro-inflammatory cytokines after I/R injury is attenuated efficiently by PPB

The level of NF-κB was increased in both the pre-ischemia and post-ischemia models and was decreased by the 4 phlorotannins, with PPB showing the strong effect (Fig. 3). The levels of TNF-α and IL-6 were also increased by I/R injury and were attenuated by adding the 4 phlorotannins, among which PPB showed the strong effect.

During I/R injury, TLR4 initiates the inflammatory response by increasing the production of NF-κB-dependent cytokines such as TNF-α [5,6]. In addition, IL-6 is released upon TLR4 activation during I/R injury and amplifies inflammation [19,20]. The RAGE pathway also induces NF-κB activation and leads to an increase in TNF-α and IL-6 levels [21]. TNF-α and IL-6 induce tubular cell death during renal I/R injury [22].

2.4. TEC death induced by I/R injury is attenuated efficiently by PPB

TEC apoptosis induced by I/R injury was attenuated by adding E. cava extracts in both the pre-ischemia and post-ischemia models.

2.5. PPB attenuated cell death signals which induced by I/R injury

The extracellular signal-regulated kinases-1 and -2 (Erk1/2), the c-Jun N-terminal kinases (SAPK/JNK), and p38 mitogen-activated protein kinases (MAPKs) are involved in cell death induced by I/R injury [23]. ERK signaling is involved in injury and apoptosis of kidney cells during I/R injury [24-26]. Both p38 and JNK pathways induce tubular cell death and their inhibition reduce apoptosis and inflammation induced by I/R injury [27,28].

In our study, these cell death signals were increased by I/R injury in TEC, and were attenuated most significantly by PPB in both the pre-ischemia and post-ischemia models.

Manuscript: Results and discussion section, Line 73 (new)

2. Results and Discussion

2.1. Attenuation of HMGB1 release from TECs after H/R injury by the phlorotannins from E. cava extracts

In this study, we used mouse kidney tubular cells (TCMK-1) as TECs. In the pre-hypoxia model, the HMGB1 level was increased by H/R injury in both TEC lysate and supernatant (Fig. 1A, B), suggesting that TECs injured by H/R increased the synthesis and release of HMGB1. HMGB1 levels in both the TEC lysate and supernatant were decreased by individual phlorotannins added before TECs were exposed to hypoxia; among individual phlorotannins, PPB and DK showed the strongest attenuation effects.

In the post-hypoxia treatment model, the HMGB1 level increased by H/R injury was also decreased by adding E. cava extracts before reperfusion in both TEC lysate and supernatant. Among the 4 phlorotannins, the effect of DK and PPB was the most significant (Fig. 1C, D).

HMGB1 is passively released in response to inflammatory stress or necrosis [18]. Our results showed that the HMGB1 release from TECs was increased after H/R injury and that this increase was attenuated by PPB most significantly among the 4 phlorotannins from E. cava.

2.2. TLR4 and RAGE expression induced by H/R injury is attenuated most efficiently by PPB

TLR4 and RAGE expression in TECs induced by H/R injury was attenuated by the 4 phlorotannins in the pre-hypoxia treatment model; among them, PPB had the strongest effect on TLR4 and RAGE expression (Fig. 2A-C, E). The patterns were the same in the post-hypoxia treatment (Fig. 2A, B, D, F).

TLR4 [19] and RAGE [20] are the primary cell membrane receptors that bind HMGB1. In H/R injury, this binding leads to pro-inflammatory signaling pathway activation by TLR and RAGE.

2.3. Expression of NF-kB and pro-inflammatory cytokines after H/R injury is attenuated efficiently by PPB

The level of NF-κB was increased in both the pre-hypoxia and post-hypoxia treatment models and was decreased by the 4 phlorotannins, with PPB showing the strong effect (Fig. 3 and Fig. S2). The levels of TNF-α and IL-6 were also increased by H/R injury and were attenuated by adding the 4 phlorotannins, among which PPB showed the strong effect.

During H/R injury, TLR4 initiates the inflammatory response by increasing the production of NF-κB-dependent cytokines such as TNF-α [5,6]. In addition, IL-6 is released upon TLR4 activation during H/R injury and amplifies inflammation [21,22]. The RAGE pathway also induces NF-κB activation and leads to an increase in TNF-α and IL-6 levels [23]. TNF-α and IL-6 induce tubular cell death during renal H/R injury [24].

2.4. TEC apoptosis induced by H/R injury is attenuated efficiently by PPB

TEC apoptosis induced by H/R injury was attenuated by adding the phlorotannin from E. cava extracts in both the pre-hypoxia and post-hypoxia treatments (Figs. 4A-C).

2.5. PPB attenuated cell death signals induced by H/R injury

The level of SAPK/JNK was increased in both the pre-hypoxia and post-hypoxia treatment models and was decreased by the 4 phlorotannins, with PPB showing the  effect (Fig. 4D, E and Fig. S3). The extracellular signal-regulated kinases-1 and -2 (Erk1/2), the c-Jun N-terminal kinases (SAPK/JNK), and p38 mitogen-activated protein kinases (MAPKs) are involved in cell death induced by H/R injury [25]. ERK signaling is involved in injury and apoptosis of kidney cells during H/R injury [26-28]. Both p38 and JNK pathways induce tubular cell death and their inhibition reduce apoptosis and inflammation induced by H/R injury [29,30].

In our study, these cell death signals were increased by H/R injury in TEC, and were attenuated most significantly by PPB in both the pre-hypoxia and post-hypoxia treatment models.

Previous one study shows that intraperitoneally administration of E. cava polyphenols at 10 mg/kg and 50 mg/kg decreased rat brain infarct size and neuronal cell apoptosis [31] and the other studies shows that 10 ug/ml or 1-50 µM only DK from E. cava have protective effects on oxidative stress-induced apoptosis in endothelial progenitor cells (EPCs) [32], primary cortical neurons and HT22 neurons [33]. However, this study tried to mimic renal ischemia reperfusion injury in vitro and validated effects on 4 phlorotannins including dieckol (DK), phlorofucofuroeckol A (PFFA), pyrogallol phloroglucinol-6,6-bieckol (PPB), or 2,7-phloroglucinol-6,6-bieckol (PHB) from E. cava.

Reviewer 2 Report

Using an experimental model of mouse renal tubular epithelial cells, TCMK-1, this study evaluated the effects of Ecklonia cava extracts against ischemia/reperfusion (I/R) injury. The authors used two different protocols for E. cava treatment. The end of introduction should be modified because the E. cava treatment in all phases in the “preischemic model” is not clear.  “Pre-ischemic method”should be modyfied in “pre-ischemic tretment”; the same for “post-ischemic method”.

The values of statistic analysis and the comparison among groups are not reported in the legends  but should be added.  Eliminate the use of the letter and the sentence of “Same letters represent no significant difference” in all figures.

Legend of Figure 1 is not clear: this legend should be rewritten with a better description of panels A, B, C and D.

Figure 3: the quantification and the statistical analysis  are absent. It’s impossible to describe the results, the same for figure 4 panel D and panel E.

Tunel staining evaluates only apoptosis , modify “death” with apoptosis.

An explanation of the E cava dose used in this study should be reported.

The discussion is quite absent as well as a comparison with the results previously published. The discussion should be completely rewritten.

Methods: A detailed description of extraction and isolation of E.cava components is absent and should be added as well as a detailed description of I/R model, for example the use of an anoxia chamber.

The results of MAPKs are not reported in the conclusion section.

Author Response

Response to Reviewer 2 Comments

Point 1: Using an experimental model of mouse renal tubular epithelial cells, TCMK-1, this study evaluated the effects of Ecklonia cava extracts against ischemia/reperfusion (I/R) injury. The authors used two different protocols for E. cava treatment. The end of introduction should be modified because the E. cava treatment in all phases in the “preischemic model” is not clear.  “Pre-ischemic method” should be modyfied in “pre-ischemic tretment”; the same for “post-ischemic method”.

Response 1: We appreciated this comment. When we looked thoroughly all manuscript, we worry about meaning confusion of “ischemia” or “ischemia/reperfusion”. So, we changed “ischemia” and “ischemia reperfusion (I/R)” to “hypoxia” and “hypoxia/reoxygenation (H/R) injury”. In additional, as above comment, we changed also “model” to “treatment” in all manuscript section.

Manuscript: title (previous)

Pyrogallol-phloroglucinol-6,6-bieckol from Ecklonia cava attenuates kidney tubular cell death in in vitro renal ischemia/reperfusion models

Manuscript: title (new)

Pyrogallol-phloroglucinol-6,6-bieckol from Ecklonia cava attenuates tubular epithelial cell (TCMK-1) death in hypoxia/reoxygenation injury

Manuscript: Abstract, Line 25 (previous)

The ischemia/reperfusion (I/R) injury causes serious complications after the blood supply to the kidney is stopped during surgery. The main mechanism of I/R injury is the release of high-mobility group protein B1 (HMGB1) from injured tubular epithelial cells (TEC), which triggers TLR4 or RAGE signaling, leading to cell death. We evaluated whether the extracts of Ecklonia cava would attenuate TEC death induced by I/R injury. We also evaluated which phlorotannin—dieckol (DK), phlorofucofuroeckol A (PFFA), pyrogallol phloroglucinol-6,6-bieckol (PPB), or 2,7-phloroglucinol-6,6-bieckol (PHB)—would have the most potent effect in the context of I/R injury. We used a pre-ischemia model, in which the E. cava extracts were added before the onset of ischemia, and a post-ischemia model, in which the extracts were added before start of reperfusion. PPB most effectively reduced HMGB1 release and the expression of TLR4 and RAGE induced by I/R injury in both pre- and post-ischemia models. PPB also most effectively inhibited expression of NF-kB and release of the inflammatory cytokines TNF-α and IL-6 in both models. PPB most effectively inhibited cell death and expression of cell death signaling molecules such as Erk/pErk, JNK/pJNK, and p38/pp38. These results suggest that PPB blocks the HGMB1–TLR4/RAGE signaling pathway and decreases TEC death induced by I/R, and that PPB can be a novel target for renal I/R injury therapy

Manuscript: Abstract, Line 25 (new)

The hypoxia/reoxygenation (H/R) injury causes serious complications after the blood supply to the kidney is stopped during surgery. The main mechanism of I/R injury is the release of high-mobility group protein B1 (HMGB1) from injured tubular epithelial cells (TEC, TCMK-1 cell), which triggers TLR4 or RAGE signaling, leading to cell death. We evaluated whether the extracts of Ecklonia cava (E. cava) would attenuate TEC death induced by H/R injury. We also evaluated which phlorotannin—dieckol (DK), phlorofucofuroeckol A (PFFA), pyrogallol phloroglucinol-6,6-bieckol (PPB), or 2,7-phloroglucinol-6,6-bieckol (PHB)—would have the most potent effect in the context of H/R injury. We used for pre-hypoxia treatment, in which the phlorotannins from E. cava extracts were added before the onset of hypoxia, and a post- hypoxia treatment, in which the phlorotannins were added before start of reperfusion. PPB most effectively reduced HMGB1 release and the expression of TLR4 and RAGE induced by H/R injury in both pre- and post- hypoxia treatment. PPB also most effectively inhibited expression of NF-kB and release of the inflammatory cytokines TNF-α and IL-6 in both models. PPB most effectively inhibited cell death and expression of cell death signaling molecules such as Erk/pErk, JNK/pJNK, and p38/pp38. These results suggest that PPB blocks the HGMB1–TLR4/RAGE signaling pathway and decreases TEC death induced by H/R, and that PPB can be a novel target for renal H/R injury therapy.

Manuscript: Introduction, Line 64 (previous)

Here, we evaluated whether extracts of E. cava would attenuate TEC death induced by I/R injury. We used two models: a pre-ischemia model, in which the E. cava extract was added before the onset of ischemia, and a post-ischemia model, in which the E. cava extract was added before the start of reperfusion. In addition, we evaluated which phlorotannin—dieckol (DK), phlorofucofuroeckol A (PFFA), pyrogallol phloroglucinol-6,6-bieckol (PPB), or 2,7-phloroglucinol-6,6-bieckol (PHB) —would have the most potent effect in the context of I/R injury.

Manuscript: Introduction, Line 64 (new)

Here, we evaluated whether phlorotannins from E. cava extract would attenuate TEC death induced by I/R injury. Commonly, renal hypoxia/reoxygenation (H/R) was established to simulate renal I/R injury in vitro [16, 17] and we also used H/R model. We used two treatment models: a pre-hypoxia model, in which the phlorotannins was added before the onset of hypoxia, and a post- hypoxia model, in which the phlorotannins was added before the start of reoxygenation. In addition, we evaluated which phlorotannin—dieckol (DK), phlorofucofuroeckol A (PFFA), pyrogallol phloroglucinol-6,6-bieckol (PPB), or 2,7-phloroglucinol-6,6-bieckol (PHB) —would have the most potent effect in the context of H/R injury.

Manuscript: Results and Discussion, Line 73 (previous)

2.1. Attenuation of HMGB1 release from TECs after I/R injury by E. cava extracts

In this study, we used mouse kidney tubular cells (TCMK-1) as TECs. In the pre-ischemia model, the HMGB1 level was increased by I/R injury in both TEC lysate and supernatant (Fig. 1A, B), suggesting that TECs injured by I/R increased the synthesis and release of HMGB1. HMGB1 levels in both the TEC lysate and supernatant were decreased by individual phlorotannins added before TECs were exposed to ischemia; among individual phlorotannins, PPB and DK showed the strongest attenuation effects.

In the post-ischemia model, the HMGB1 level increased by I/R injury was also decreased by adding E. cava extracts before reperfusion in both TEC lysate and supernatant. Among the 4 phlorotannins, the effect of DK and PPB was the  significant (Fig. 1C, D). HMGB1 is passively released in response to inflammatory stress or necrosis [16]. Our results showed that the HMGB1 release from TECs was increased after I/R injury and that this increase was attenuated by PPB most significantly among the 4 E. cava phlorotannins.

Manuscript: Results and Discussion, Line 73 (new)

2.1. Attenuation of HMGB1 release from TECs after H/R injury by the phlorotannins from E. cava extracts

In this study, we used mouse kidney tubular cells (TCMK-1) as TECs. In the pre-hypoxia model, the HMGB1 level was increased by H/R injury in both TEC lysate and supernatant (Fig. 1A, B), suggesting that TECs injured by H/R increased the synthesis and release of HMGB1. HMGB1 levels in both the TEC lysate and supernatant were decreased by individual phlorotannins added before TECs were exposed to hypoxia; among individual phlorotannins, PPB and DK showed the strongest attenuation effects.

In the post-hypoxia treatment model, the HMGB1 level increased by H/R injury was also decreased by adding E. cava extracts before reperfusion in both TEC lysate and supernatant. Among the 4 phlorotannins, the effect of DK and PPB was the most significant (Fig. 1C, D). HMGB1 is passively released in response to inflammatory stress or necrosis [18]. Our results showed that the HMGB1 release from TECs was increased after H/R injury and that this increase was attenuated by PPB most significantly among the 4 phlorotannins from E. cava.

Manuscript: Results and Discussion, Line 104 (previous)

2.2. TLR4 and RAGE expression induced by I/R injury is attenuated most efficiently by PPB

TLR4 and RAGE expression in TECs induced by I/R injury was attenuated by the 4 phlorotannins in the pre-ischemia model; among them, PPB had the strongest effect on TLR4 and RAGE expression (Fig. 2A, B, C, E). The patterns were the same in the post-ischemia model (Fig. 2A, B, D, F).

TLR4 [17] and RAGE [18] are the primary cell membrane receptors that bind HMGB1. In I/R injury, this binding leads to pro-inflammatory signaling pathway activation by TLR and RAGE.

Manuscript: Results and Discussion, Line 104 (new)

2.2. TLR4 and RAGE expression induced by H/R injury is attenuated most efficiently by PPB

TLR4 and RAGE expression in TECs induced by H/R injury was attenuated by the 4 phlorotannins in the pre-hypoxia treatment model; among them, PPB had the strongest effect on TLR4 and RAGE expression (Fig. 2A-C, E). The patterns were the same in the post-hypoxia treatment (Fig. 2A, B, D, F).

TLR4 [19] and RAGE [20] are the primary cell membrane receptors that bind HMGB1. In H/R injury, this binding leads to pro-inflammatory signaling pathway activation by TLR and RAGE.

Manuscript: Results and Discussion, Line 126 (previous)

2.3. Expression of NF-kB and pro-inflammatory cytokines after I/R injury is attenuated efficiently by PPB

The level of NF-κB was increased in both the pre-ischemia and post-ischemia models and was decreased by the 4 phlorotannins, with PPB showing the strong effect (Fig. 3). The levels of TNF-α and IL-6 were also increased by I/R injury and were attenuated by adding the 4 phlorotannins, among which PPB showed the strong effect.

During I/R injury, TLR4 initiates the inflammatory response by increasing the production of NF-κB-dependent cytokines such as TNF-α [5,6]. In addition, IL-6 is released upon TLR4 activation during I/R injury and amplifies inflammation [19,20]. The RAGE pathway also induces NF-κB activation and leads to an increase in TNF-α and IL-6 levels [21]. TNF-α and IL-6 induce tubular cell death during renal I/R injury [22].

Manuscript: Results and Discussion, Line 126 (new)

2.3. Expression of NF-kB and pro-inflammatory cytokines after H/R injury is attenuated efficiently by PPB

The level of NF-κB was increased in both the pre-hypoxia and post-hypoxia treatment models and was decreased by the 4 phlorotannins, with PPB showing the strong effect (Fig. 3). The levels of TNF-α and IL-6 were also increased by H/R injury and were attenuated by adding the 4 phlorotannins, among which PPB showed the strong effect.

During H/R injury, TLR4 initiates the inflammatory response by increasing the production of NF-κB-dependent cytokines such as TNF-α [5,6]. In addition, IL-6 is released upon TLR4 activation during H/R injury and amplifies inflammation [21,22]. The RAGE pathway also induces NF-κB activation and leads to an increase in TNF-α and IL-6 levels [23]. TNF-α and IL-6 induce tubular cell death during renal H/R injury [24].

Manuscript: Results and Discussion, Line 149 (previous)

2.4. TEC death induced by I/R injury is attenuated efficiently by PPB

TEC apoptosis induced by I/R injury was attenuated by adding E. cava extracts in both the pre-ischemia and post-ischemia models.

Manuscript: Results and Discussion, Line 149 (new)

2.4. TEC apoptosis induced by H/R injury is attenuated efficiently by PPB

TEC apoptosis induced by H/R injury was attenuated by adding the phlorotannin from E. cava extracts in both the pre-hypoxia and post-hypoxia treatments.

Manuscript: Results and Discussion, Line 166 (previous)

2.5. PPB attenuated cell death signals which induced by I/R injury

The extracellular signal-regulated kinases-1 and -2 (Erk1/2), the c-Jun N-terminal kinases (SAPK/JNK), and p38 mitogen-activated protein kinases (MAPKs) are involved in cell death induced by I/R injury [23]. ERK signaling is involved in injury and apoptosis of kidney cells during I/R injury [24-26]. Both p38 and JNK pathways induce tubular cell death and their inhibition reduce apoptosis and inflammation induced by I/R injury [27,28].

In our study, these cell death signals were increased by I/R injury in TEC, and were attenuated most significantly by PPB in both the pre-ischemia and post-ischemia models.

Manuscript: Results and Discussion, Line 166 (new)

2.5. PPB attenuated cell death signals induced by H/R injury

The extracellular signal-regulated kinases-1 and -2 (Erk1/2), the c-Jun N-terminal kinases (SAPK/JNK), and p38 mitogen-activated protein kinases (MAPKs) are involved in cell death induced by H/R injury [25]. ERK signaling is involved in injury and apoptosis of kidney cells during H/R injury [26-28]. Both p38 and JNK pathways induce tubular cell death and their inhibition reduce apoptosis and inflammation induced by H/R injury [29,30].

In our study, these cell death signals were increased by H/R injury in TEC, and were attenuated most significantly by PPB in both the pre-hypoxia and post-hypoxia treatment models.

Manuscript: Results and Discussion, Line 176 (previous)

3.2. Kidney ischemia reperfusion cell models

To examine the inhibitory effects of 4 phlorotannins DK, PHB, PFFA and PPB from E. cava extract, we designed two types of ischemia/reperfusion cell models using TCMK-1 cells; the experimental scheme is shown in Figure S1.

3.2.1. Pre-ischemia model

Four prepared phlorotannins (2.5 μg/ml) were added to growth medium, which was used to treat TCMK-1 cells for 6 h before ischemia. After treatment, the medium was completely removed, and the cells were washed thoroughly with PBS. Ischemia medium contained 0.5% FBS and 1% penicillin/streptomycin in DMEM. The TCMK-1 cells were cultured in ischemia medium with the 4 phlorotannins and exposed to a mixture of 1% O2, 5% CO2, 94% N2 for 14 h. After incubation, the ischemia medium was removed, and the cells were washed with PBS. The cells were cultured in growth medium containing 4 phlorotannins in a 5% CO2 incubator for 9 h, and then used for experiments.

3.2.2. Post-ischemia model

TCMK-1 cells were cultured in growth medium for 6 h before ischemia, growth medium was completely removed, and the cells were washed with PBS. Ischemia treatment was performed as in 3.2.1. After the treatment, the ischemia medium was removed, and the cells were washed with PBS. The cells were cultured in growth medium containing 4 phlorotannins in a 5% CO2 incubator for 9 h, and then used for experiments.

Manuscript: Materials and methods, Line 176 (new)

3.2. Hypoxia/reoxygenation injury cell models

To examine the inhibitory effects of 4 phlorotannins DK, PHB, PFFA and PPB from E. cava extract, we designed two types of H/R cell models using TCMK-1 cells; the experimental method and image scheme is shown in Figure S1.

3.2.1. Pre-hypoxia treatment model

Four prepared phlorotannins (2.5 μg/ml) were added to growth medium, which was used to treat TCMK-1 cells for 6 h before hypoxia. After treatment, the medium was completely removed, and the cells were washed thoroughly with PBS. Hypoxia medium contained 0.5% FBS and 1% penicillin/streptomycin in DMEM. The TCMK-1 cells were cultured in hypoxia medium with the 4 phlorotannins and exposed to a mixture of 1% O2, 5% CO2, 94% N2 for 14 h. After incubation, the hypoxia medium was removed, and the cells were washed with PBS. The cells were cultured with fresh oxygenated medium containing FBS, 1% penicillin/streptomycin, 4 phlorotannins in a 5% CO2 incubator for 9 h, and then used for experiments.

3.2.2. Post-hypoxia treatment model

TCMK-1 cells were cultured in growth medium for 6 h before hypoxia, growth medium was completely removed, and the cells were washed with PBS. Hypoxia exposure was performed as in 3.2.1. Pre-hypoxia treatment model of material and methods section. After the hypoxia treatment, the hypoxia chamber was dismantled and the hypoxia medium was removed, and the cells were washed with PBS. The cells were cultured with fresh oxygenated medium containing FBS, 1% penicillin/streptomycin, 4 phlorotannins in a mixture of 20% O2, 5% CO2, 75% N2 for 9 h, and then used for experiments. The experimental method and image scheme is shown in Figure S1.

Point 2: The values of statistic analysis and the comparison among groups are not reported in the legends but should be added. Eliminate the use of the letter and the sentence of “Same letters represent no significant difference” in all figures.

Response 2-1: We appreciated this comment. The statistical techniques used in this study were compared the statistical difference among 6 groups using the non-parametric Kruskal-Wallis test, and the difference between 2 groups was compared using the Mann-Whitney U test for post-test. For example, If there are significant (P ≤ 0.05) between 2 groups, we marked different character but, there are no significant (P ≥ 0.05) between 2 groups, we marked same character (below left image). As this comment, we think our figure and manuscript can be confusing. Therefore, we changed all previous figures to new figures with some special symbols (*, #, $; below right image) in manuscript.

In additional, as this comment, we removed sentence of “Same letters represent no significant difference” in all figures and we marked that * (asterisk) compared with normoxia PBS (PBS), $ compared with hypoxia PBS (Hx/PBS) and # compared with hypoxia/PBS (Hx/PPB; Figure legend section, Line 92, 114, 155).

Manuscript: Figure 1 legend, Line 92 (previous)

Figure 1. Inhibitory effects of phlorotannins from E. cava extract on HMGB1 synthesis and secretion in pre-ischemia and post-ischemia models (A, B) To examine the preventive effects of 4 phlorotannins from E. cava extract (DK, PHB, PPB, PFFA), they were added to mouse kidney tubular cells (TCMK-1) before ischemia (pre-ischemia model). (C, D) To examine the therapeutic effects of the 4 phlorotannins, they were added to TCMK-1 after ischemia (post-ischemia model). HMGB1 (A, C) synthesis and (B, D) secretion levels were measured by ELISA in the TCMK-1 lysate and supernatant, respectively. All levels are normalized to those in cells treated with PBS under normoxic control conditions. Same letters represent no significant difference (p ≥ 0.05). DK, dieckol; PHB, 2,7-phloroglucinol-6,6-bieckol, PPB, pyrogallol-phloroglucinol-6,6-bieckol; PFFA, phlorofucofuroeckol A; HMGB1, high mobility group box 1.

Manuscript: Figure 1 & legend, Line 92 (new)

Figure 1. Inhibitory effects of phlorotannins from E. cava extract on HMGB1 synthesis and secretion in pre-hypoxia and post-hypoxia treatment (A, B) To examine the preventive effects of 4 phlorotannins from E. cava extract (DK, PHB, PPB, PFFA), they were added to mouse kidney tubular cells (TCMK-1) before hypoxia (pre-hypoxia model). (C, D) To examine the therapeutic effects of the 4 phlorotannins, they were added to TCMK-1 after hypoxia (post- hypoxia model). HMGB1 (A, C) synthesis and (B, D) secretion levels were measured by ELISA in the TCMK-1 lysate and supernatant, respectively. All levels are normalized to those in cells treated with PBS under normoxic control conditions. Significance represented as; *, <0.05 versus PBS; $, <0.05 versus Hx/PBS; #, < 0.05 versus Hx/PPB. DK, dieckol; PHB, 2,7-phloroglucinol-6,6-bieckol, PPB, pyrogallol-phloroglucinol-6,6-bieckol; PFFA, phlorofucofuroeckol A; HMGB1, high mobility group box 1.

Manuscript: Figure 2 legend, Line 114 (previous)

Figure 2. Inhibitory effects of phlorotannins from E. cava extract on TLR4 and RAGE expression in pre-ischemia and post-ischemia models Mouse kidney tubular cells (TCMK-1) were treated with 4 phlorotannins (DK, PFFA, PPB, PHB) before ischemia (pre-ischemia model) or after ischemia (post-ischemia model). (A, B) Microscopic fluorescence images showing (A) TLR4 and (B) RAGE protein expression (green) in the pre-ischemia model (upper rows) and post-ischemia model (bottom rows). Nuclei were stained with DAPI (blue). Scale bar = 50 µm. (C-F) Quantification of (C, D) TLR4 and (E, F) RAGE expression using representative images from (A) and (B) in the pre-ischemia model and post-ischemia model. Same letters represent no significant difference (p ≥ 0.05). DK, dieckol; PHB, 2,7-phloroglucinol-6,6-bieckol, PFFA, phlorofucofuroeckol A; PPB, pyrogallol-phloroglucinol-6,6-bieckol; TLR4, Toll-like receptor 4; RAGE, receptor for advanced glycation end-products.

Manuscript: Figure 2 & legend, Line 114 (new)

Figure 2. Inhibitory effects of phlorotannins from E. cava extract on TLR4 and RAGE expression in pre-hypoxia and post-hypoxia treatment using Mouse kidney tubular cells (TCMK-1) were treated with 4 phlorotannins (DK, PFFA, PPB, PHB) before hypoxia (pre-hypoxia model) or after hypoxia (post-hypoxia model). (A, B) Microscopic fluorescence images showing (A) TLR4 and (B) RAGE protein expression (green) in the pre-hypoxia treatment (upper rows) and post- hypoxia treatment (bottom rows). Nuclei were stained with DAPI (blue). Scale bar = 50 µm. (C-F) Quantification of (C, D) TLR4 and (E, F) RAGE expression using representative images from (A) and (B) in the pre-hypoxia treatment model and post-hypoxia treatment. Significance represented as; **, <0.01 versus PBS; $$, <0.01 versus Hx/PBS; #, < 0.05, ##, < 0.01 versus Hx/PPB. DK, dieckol; PHB, 2,7-phloroglucinol-6,6-bieckol, PFFA, phlorofucofuroeckol A; PPB, pyrogallol-phloroglucinol-6,6-bieckol; TLR4, Toll-like receptor 4; RAGE, receptor for advanced glycation end-products.

Manuscript: Figure 4 legend, Line 155 (previous)

Figure 4. Inhibitory effects of phlorotannins from E. cava extract on cell death and cell death–related MAPK pathway molecules in pre-ischemia and post-ischemia models Mouse kidney tubular cells (TCMK-1) were treated with the 4 phlorotannins (DK, PHB, PPB, PFFA) before ischemia (pre-ischemia model) or after ischemia (post-ischemia model). (A) Microscopic fluorescence images showing TUNEL-positive apoptotic cells (pink) and nuclei (blue, DAPI) in the pre-ischemia model (upper row) and post-ischemia model (bottom row). Scale bar = 50 µm. Quantification of TUNEL-positive apoptotic cells using representative images is shown in (B) for the pre-ischemia model and in (C) for the post-ischemia model. Same letters represent no significant difference (p ≥ 0.05). Apoptosis-related MAPK pathway molecules were detected by immunostaining in (D) pre-ischemia model and (E) post-ischemia model. DK, dieckol; PHB, 2,7-phloroglucinol-6,6-bieckol; PFFA, phlorofucofuroeckol A; PPB, pyrogallol-phloroglucinol-6,6-bieckol; TUNEL, terminal deoxynucleotidyl transferase dUTP nick end labeling.

Manuscript: Figure 4 legend, Line 155 (new)

Figure 4. Inhibitory effects of phlorotannins from E. cava extract on apoptosis and apoptotic cell death–related MAPK pathway molecules in pre-hypoxia and post-hypoxia treatment Mouse kidney tubular cells (TCMK-1) were treated with the 4 phlorotannins (DK, PHB, PPB, PFFA) before hypoxia (pre-hypoxia treatment) or after hypoxia (post-hypoxia treatment). (A) Microscopic fluorescence images showing TUNEL-positive apoptotic cells (pink) and nuclei (blue, DAPI) in the pre-hypoxia treatment (upper row) and post-hypoxia treatment (bottom row). Scale bar = 50 µm. Quantification of TUNEL-positive apoptotic cells using representative images is shown in (B) for the pre-hypoxia treatment and in (C) for the post-hypoxia treatment. Significance represented as; *, <0.05 versus PBS; $, <0.05 versus Hx/PBS; #, < 0.05 versus Hx/PPB. Apoptosis-related MAPK pathway molecules were detected by immunostaining in (D) pre-hypoxia treatment and (E) post-hypoxia treatment. DK, dieckol; PHB, 2,7-phloroglucinol-6,6-bieckol; PFFA, phlorofucofuroeckol A; PPB, pyrogallol-phloroglucinol-6,6-bieckol; TUNEL, terminal deoxynucleotidyl transferase dUTP nick end labeling.

Response 2-2: We also changed statistical analysis and the comparison in materials and methods section in manuscript. (Materials and methods section, Line 261 in manuscript).

Manuscript: Materials and methods, Line 261 (previous)

3.7. Statistical analysis

The Kruskal–Wallis test and the Mann–Whitney U post-hoc test were used to determine the significance of differences among the normoxia control, ischemia/reperfusion, and ischemia/reperfusion with phlorotannins groups. Results are presented as mean ± SD, and statistical significance was accepted for p values <0.05. The analysis was performed using SPSS version 22 (IBM Corporation, NY, USA). Same letters represent no significant difference (p ≥ 0.05).

Manuscript: Materials and methods, Line 261 (new)

3.8. Statistical analysis

The statistical analysis was conducted using SPSS version 22 software (IBM Co.; NY, USA). In this study, statistical differences were compared among 6 groups (PBS, Hx/PBS, Hx/DK, Hx/PHB, Hx/PPB, Hx/PFFA) in pre-hypoxia and post-hypoxia treatment using the non-parametric Kruskal-Wallis test. For post-test, difference between 2 groups was compared using the Mann-Whitney U test. All experiments were performed in triplicate, and results are presented as means ± standard deviation. * (asterisk) means comparing with PBS, $ comparing with Hx/PBS and # comparing  with Hx/PPB.

Point 3: Legend of Figure 1 is not clear: this legend should be rewritten with a better description of panels A, B, C and D.

Response 3: We appreciated this comment. As you’ve commented, newly written for the reader to understand. This legend was changed and added in manuscript (Figure legend section, Line 88 in manuscript).

Manuscript: Figure legend, Line 88 (previous)

Figure 1. Inhibitory effects of phlorotannins from E. cava extract on HMGB1 synthesis and secretion in pre-ischemia and post-ischemia models (A, B) To examine the preventive effects of 4 phlorotannins from E. cava extract (DK, PHB, PPB, PFFA), they were added to mouse kidney tubular cells (TCMK-1) before ischemia (pre-ischemia model). (C, D) To examine the therapeutic effects of the 4 phlorotannins, they were added to TCMK-1 after ischemia (post-ischemia model). HMGB1 (A, C) synthesis and (B, D) secretion levels were measured by ELISA in the TCMK-1 lysate and supernatant, respectively. All levels are normalized to those in cells treated with PBS under normoxic control conditions. Same letters represent no significant difference (p ≥ 0.05). DK, dieckol; PHB, 2,7-phloroglucinol-6,6-bieckol, PPB, pyrogallol-phloroglucinol-6,6-bieckol; PFFA, phlorofucofuroeckol A; HMGB1, high mobility group box 1.

Manuscript: Figure legend, Line 88 (new)

Figure 1. Inhibitory effects of phlorotannins from E. cava extract on HMGB1 synthesis and secretion in pre-hypoxia and post-hypoxia treatment (A, B) To examine the preventive effects of 4 phlorotannins from E. cava extract (DK, PHB, PPB, PFFA), they were added to mouse kidney tubular cells (TCMK-1) before hypoxia (pre-hypoxia treatment). (C, D) To examine the therapeutic effects of the 4 phlorotannins, they were added to TCMK-1 after hypoxia (post-hypoxia treatment). In each treatment model, (A, C) HMGB1 synthesis in cell lysate and (B, D) secretion levels in cell culture medium were measured by ELISA. All levels are normalized to those in cells treated with PBS under normoxic control conditions. Significance represented as; *, <0.05 versus PBS; $, <0.05 versus Hx/PBS; #, < 0.05 versus Hx/PPB. DK, dieckol; PHB, 2,7-phloroglucinol-6,6-bieckol, PPB, pyrogallol-phloroglucinol-6,6-bieckol; PFFA, phlorofucofuroeckol A; HMGB1, high mobility group box 1.

Point 4: Figure 3: the quantification and the statistical analysis are absent. It’s impossible to describe the results, the same for figure 4 panel D and panel E.

Response 4: We appreciated this comment. As you’ve commented, quantification and the statistical analysis of immunoblotting result (Figure 3 and Figure 4D, 4E) are absent. All immunoblotting results are quantified and statistically analyzed. The quantification results added in supplementary figure 2 and 3 (Figure S2, 3) and described in manuscript (Results and Discussion section, Line 170 in manuscript).

Manuscript: Supplementary figure 2 (new)

Inflammatory factors including NF-κB, TNF-α and IL-6 expression are quantified from immunoblotting expression in (A) pre-hypoxia and (B) post-hypoxia treatment model. All levels are normalized to those in cells treated with PBS. Significance represented as; *, <0.05, **, <0.01 versus PBS; $, <0.05, $$, <0.01 versus Hx/PBS; #, < 0.05, ##, <0.01 versus Hx/PPB.

Manuscript: Results and discussion, Line 127 (new)

2.3. Expression of NF-kB and pro-inflammatory cytokines after H/R injury is attenuated efficiently by PPB

The level of NF-κB was increased in both the pre-hypoxia and post-hypoxia treatment models and was decreased by the 4 phlorotannins, with PPB showing the strong effect (Fig. 3 and Fig. S2). The levels of TNF-α and IL-6 were also increased by H/R injury and were attenuated by adding the 4 phlorotannins, among which PPB showed the strong effect.

Manuscript: Supplementary figure 3 (new)

Apoptosis-related phosphorylated MAPKs including pERK1/2, pSAPK/JNK and pP38 expression are quantified using non-phosphorylated protein expression including ERK1/2, SAPK/JNK and pP38 in (A) pre-hypoxia and (B) post-hypoxia treatment model. All levels are normalized to those in cells treated with PBS. Significance represented as; **, <0.01 versus PBS; $, <0.05, $$, <0.01 versus Hx/PBS; #, < 0.05, ##, <0.01 versus Hx/PPB.

Manuscript: Results and discussion, Line 170 (new)

2.5. PPB attenuated cell death signals induced by H/R injury

The level of SAPK/JNK was increased in both the pre-hypoxia and post-hypoxia treatment models and was decreased by the 4 phlorotannins, with PPB showing the  effect (Fig. 4D, E and Fig. S3). The extracellular signal-regulated kinases-1 and -2 (Erk1/2), the c-Jun N-terminal kinases (SAPK/JNK), and p38 mitogen-activated protein kinases (MAPKs) are involved in cell death induced by H/R injury [25]. ERK signaling is involved in injury and apoptosis of kidney cells during H/R injury [26-28]. Both p38 and JNK pathways induce tubular cell death and their inhibition reduce apoptosis and inflammation induced by H/R injury [29,30].

In our study, these cell death signals were increased by H/R injury in TEC, and were attenuated most significantly by PPB in both the pre-hypoxia and post-hypoxia treatment models.

Point 5: Tunel staining evaluates only apoptosis , modify “death” with apoptosis.

Response 5: We appreciated this comment. As you’ve commented, TUNEL assay can evaluate only apoptotic cell death. So, we changed “death” to “apoptosis” or “apoptotic cell death” in manuscript (Results and Discussion section, Line 143 and Figure legend, Line 147 in manuscript).

Manuscript: Results and discussion, Line 143 (previous)

2.4. TEC death induced by I/R injury is attenuated efficiently by PPB

TEC apoptosis induced by I/R injury was attenuated by adding E. cava extracts in both the pre-ischemia and post-ischemia models.

Manuscript: Results and discussion, Line 143 (new)

2.4. TEC apoptosis induced by I/R injury is attenuated efficiently by PPB

TEC apoptosis induced by I/R injury was attenuated by adding E. cava extracts in both the pre-ischemia and post-ischemia treatments.

Manuscript: Figure legend, Line 147 (previous)

Figure 4. Inhibitory effects of phlorotannins from E. cava extract on cell death and cell death–related MAPK pathway molecules in pre-ischemia and post-ischemia models Mouse kidney tubular cells (TCMK-1) were treated with the 4 phlorotannins (DK, PHB, PPB, PFFA) before ischemia (pre-ischemia model) or after ischemia (post-ischemia model). (A) Microscopic fluorescence images showing TUNEL-positive apoptotic cells (pink) and nuclei (blue, DAPI) in the pre-ischemia model (upper row) and post-ischemia model (bottom row). Scale bar = 50 µm. Quantification of TUNEL-positive apoptotic cells using representative images is shown in (B) for the pre-ischemia model and in (C) for the post-ischemia model. Same letters represent no significant difference (p ≥ 0.05). Apoptosis-related MAPK pathway molecules were detected by immunostaining in (D) pre-ischemia model and (E) post-ischemia model. DK, dieckol; PHB, 2,7-phloroglucinol-6,6-bieckol; PFFA, phlorofucofuroeckol A; PPB, pyrogallol-phloroglucinol-6,6-bieckol; TUNEL, terminal deoxynucleotidyl transferase dUTP nick end labeling.

Manuscript: Figure legend, Line 147 (new)

Figure 4. Inhibitory effects of phlorotannins from E. cava extract on apoptosis and apoptotic cell death–related MAPK pathway molecules in pre-ischemia and post-ischemia treatment Mouse kidney tubular cells (TCMK-1) were treated with the 4 phlorotannins (DK, PHB, PPB, PFFA) before ischemia (pre-ischemia treatment) or after ischemia (post-ischemia treatment). (A) Microscopic fluorescence images showing TUNEL-positive apoptotic cells (pink) and nuclei (blue, DAPI) in the pre-ischemia treatment (upper row) and post-ischemia treatment (bottom row). Scale bar = 50 µm. Quantification of TUNEL-positive apoptotic cells using representative images is shown in (B) for the pre-ischemia treatment and in (C) for the post-ischemia treatment. Same letters represent no significant difference (p ≥ 0.05). Apoptosis-related MAPK pathway molecules were detected by immunostaining in (D) pre-ischemia treatment and (E) post-ischemia treatment. DK, dieckol; PHB, 2,7-phloroglucinol-6,6-bieckol; PFFA, phlorofucofuroeckol A; PPB, pyrogallol-phloroglucinol-6,6-bieckol; TUNEL, terminal deoxynucleotidyl transferase dUTP nick end labeling.

Point 6: An explanation of the E cava dose used in this study should be reported.

Response 6: We appreciated this comment. In this study, we dose not used E. cava extract and we just used four phlorotannins (DK, PHB, PPB, PFFA) from E. cava extract. When we looked thoroughly all manuscript, we found that few miswriting ‘E. cava extract’ instead of phlorotannins and that it could confuse the reader. So, we changed ‘E. cava extract’ to ‘phlorotannins’ in all manuscript section (Abstract, Introduction section, Line 63, Results and Discussion section, Line 71 and 143).

Manuscript: Abstract (previous)

The ischemia/reperfusion (I/R) injury causes serious complications after the blood supply to the kidney is stopped during surgery. The main mechanism of I/R injury is the release of high-mobility group protein B1 (HMGB1) from injured tubular epithelial cells (TEC), which triggers TLR4 or RAGE signaling, leading to cell death. We evaluated whether the extracts of Ecklonia cava would attenuate TEC death induced by I/R injury. We also evaluated which phlorotannin—dieckol (DK), phlorofucofuroeckol A (PFFA), pyrogallol phloroglucinol-6,6-bieckol (PPB), or 2,7-phloroglucinol-6,6-bieckol (PHB)—would have the most potent effect in the context of I/R injury. We used a pre-ischemia model, in which the E. cava extracts were added before the onset of ischemia, and a post-ischemia model, in which the extracts were added before start of reperfusion. PPB most effectively reduced HMGB1 release and the expression of TLR4 and RAGE induced by I/R injury in both pre- and post-ischemia models. PPB also most effectively inhibited expression of NF-kB and release of the inflammatory cytokines TNF-α and IL-6 in both models. PPB most effectively inhibited cell death and expression of cell death signaling molecules such as Erk/pErk, JNK/pJNK, and p38/pp38. These results suggest that PPB blocks the HGMB1–TLR4/RAGE signaling pathway and decreases TEC death induced by I/R, and that PPB can be a novel target for renal I/R injury therapy.

Manuscript: Abstract (new)

The ischemia/reperfusion (I/R) injury causes serious complications after the blood supply to the kidney is stopped during surgery. The main mechanism of I/R injury is the release of high-mobility group protein B1 (HMGB1) from injured tubular epithelial cells (TEC), which triggers TLR4 or RAGE signaling, leading to cell death. We evaluated whether the extracts of Ecklonia cava would attenuate TEC death induced by I/R injury. We also evaluated which phlorotannin—dieckol (DK), phlorofucofuroeckol A (PFFA), pyrogallol phloroglucinol-6,6-bieckol (PPB), or 2,7-phloroglucinol-6,6-bieckol (PHB)—would have the most potent effect in the context of I/R injury. We used a pre-ischemia model, in which the phlorotannins from E. cava extracts were added before the onset of ischemia, and a post-ischemia model, in which the phlorotannins were added before start of reperfusion. PPB most effectively reduced HMGB1 release and the expression of TLR4 and RAGE induced by I/R injury in both pre- and post-ischemia models. PPB also most effectively inhibited expression of NF-kB and release of the inflammatory cytokines TNF-α and IL-6 in both models. PPB most effectively inhibited cell death and expression of cell death signaling molecules such as Erk/pErk, JNK/pJNK, and p38/pp38. These results suggest that PPB blocks the HGMB1–TLR4/RAGE signaling pathway and decreases TEC death induced by I/R, and that PPB can be a novel target for renal I/R injury therapy.

Manuscript: Introduction section, Line 63 (previous)

Here, we evaluated whether extracts of E. cava would attenuate TEC death induced by I/R injury. We used two models: a pre-ischemia model, in which the E. cava extract was added before the onset of ischemia, and a post-ischemia model, in which the E. cava extract was added before the start of reperfusion. In addition, we evaluated which phlorotannin—dieckol (DK), phlorofucofuroeckol A (PFFA), pyrogallol phloroglucinol-6,6-bieckol (PPB), or 2,7-phloroglucinol-6,6-bieckol (PHB) —would have the most potent effect in the context of I/R injury.

Manuscript: Introduction section, Line 63 (new)

Here, we evaluated whether phlorotannins from E. cava extract would attenuate TEC death induced by I/R injury. We used two models: a pre-ischemia model, in which the phlorotannins was added before the onset of ischemia, and a post-ischemia model, in which the phlorotannins was added before the start of reperfusion. In addition, we evaluated which phlorotannin—dieckol (DK), phlorofucofuroeckol A (PFFA), pyrogallol phloroglucinol-6,6-bieckol (PPB), or 2,7-phloroglucinol-6,6-bieckol (PHB) —would have the most potent effect in the context of I/R injury.

Manuscript: Results and Discussion section, Line 71 (previous)

2.1. Attenuation of HMGB1 release from TECs after I/R injury by E. cava extracts

In this study, we used mouse kidney tubular cells (TCMK-1) as TECs. In the pre-ischemia model, the HMGB1 level was increased by I/R injury in both TEC lysate and supernatant (Fig. 1A, B), suggesting that TECs injured by I/R increased the synthesis and release of HMGB1. HMGB1 levels in both the TEC lysate and supernatant were decreased by individual phlorotannins added before TECs were exposed to ischemia; among individual phlorotannins, PPB and DK showed the strongest attenuation effects.

Manuscript: Results and Discussion section, Line 71 (new)

2.1. Attenuation of HMGB1 release from TECs after I/R injury by the phlorotannins from E. cava extracts

In this study, we used mouse kidney tubular cells (TCMK-1) as TECs. In the pre-ischemia model, the HMGB1 level was increased by I/R injury in both TEC lysate and supernatant (Fig. 1A, B), suggesting that TECs injured by I/R increased the synthesis and release of HMGB1. HMGB1 levels in both the TEC lysate and supernatant were decreased by individual phlorotannins added before TECs were exposed to ischemia; among individual phlorotannins, PPB and DK showed the strongest attenuation effects.

Manuscript: Results and Discussion section, Line 143 (previous)

2.4. TEC death induced by I/R injury is attenuated efficiently by PPB

TEC apoptosis induced by I/R injury was attenuated by adding E. cava extracts in both the pre-ischemia and post-ischemia models.

Manuscript: Results and Discussion section, Line 143 (new)

2.4. TEC death induced by I/R injury is attenuated efficiently by PPB

TEC apoptosis induced by I/R injury was attenuated by adding the phlorotannin from E. cava extracts in both the pre-ischemia and post-ischemia models.

Point 7: The discussion is quite absent as well as a comparison with the results previously published. The discussion should be completely rewritten.

Response 7: We appreciated this comment. As this comment, we compared with previously published researches. There are 3 research papers about effects on E. cava extract or phlorotannins from E. cava in ischemia condition but, there are no ischemia reperfusion or hypoxia/reoxygenation injury.

One study shows that intraperitoneally administration of E. cava polyphenols at 10 mg/kg and 50 mg/kg decreased rat brain infarct size and neuronal cell apoptosis [Ref: Anat Cell Biol. 2012 Jun;45(2):103-13.]. The other studies show that 10 ug/ml or 1-50 µM dieckol from E. cava have protective effects on oxidative stress-induced apoptosis in endothelial progenitor cells (EPCs) [Ref: Free Radic Res. 2013 Jul;47(6-7):526-34.], primary cortical neurons and HT22 neurons [Ref: Korean J Physiol Pharmacol. 2019 Mar;23(2):121-130.].

However, there are few differences between ours and exist previous studies. In previous in vitro study, they just used a oxidative stress-induced damage model not ischemia reperfusion or hypoxia/reoxygenation injury and they showing only therapeutic possibilities. Despite the limitations of in vitro experiments, we used general hypoxia/reoxygenation methods [Ref: J Pharmacol Sci. 2018 Jun;137(2):170-176 & Cell Death Dis. 2018 Mar 1;9(3):338.] to mimic renal ischemia reperfusion injury in vivo. As well as experimental model, we validated effects of 4 phlorotannins against hypoxia/reoxygenation (H/R) injury.  We made pre-ischemia and post-ischemia treatment model and 4 phlorotannins was treated each model. We rewritten discussion section and added previous studies as reference papers in manuscript (Results and Discussion section, Line 181 and References in manuscript).

Manuscript: Results and Discussion, Line 181 (new)

Previous one study shows that intraperitoneally administration of E. cava polyphenols at 10 mg/kg and 50 mg/kg decreased rat brain infarct size and neuronal cell apoptosis [31] and the other studies shows that 10 ug/ml or 1-50 µM only DK from E. cava have protective effects on oxidative stress-induced apoptosis in endothelial progenitor cells (EPCs) [32], primary cortical neurons and HT22 neurons [33]. However, this study tried to mimic renal ischemia reperfusion injury in vitro and validated effects on 4 phlorotannins including dieckol (DK), phlorofucofuroeckol A (PFFA), pyrogallol phloroglucinol-6,6-bieckol (PPB), or 2,7-phloroglucinol-6,6-bieckol (PHB) from E. cava.

Manuscript: References (new)

31.     Kim, J.H.; Lee, N.S.; Jeong, Y.G.; Lee, J.H.; Kim, E.J.; Han, S.Y. Protective efficacy of an Ecklonia cava extract used to treat transient focal ischemia of the rat brain. Anat. Cell Biol. 2012, 45, 103-113.

32.     Lee, S.H.; Kim, J.Y.; Yoo, S.Y.; Kwon, S.M. Cytoprotective effect of dieckol on human endothelial progenitor cells (hEPCs) from oxidative stress-induced apoptosis. Free Radic. Res. 2013, 47, 526-34.

33.   Cui, Y.; Amarsanaa, K.; Lee, J.H.; Rhim, J.K.; Kwon, J.M.; Kim, S.H.; Park, J.M.7; Jung, S.C.;  Eun SY. Neuroprotective mechanisms of dieckol against glutamate toxicity through reactive oxygen species scavenging and nuclear factor-like 2/heme oxygenase-1 pathway. Korean J. Physiol. Pharmacol. 2019, 23, 121-130

Point 8: Methods: A detailed description of extraction and isolation of E.cava components is absent and should be added as well as a detailed description of I/R model, for example the use of an anoxia chamber.

Response 8-1: We appreciated this comment. As this comment,  detailed descriptions of extraction and isolation of E.cava components methods are poor in manuscript. So, we added detailed description of isolation of E. cava method in material and methods section of manuscript (Material and methods section, Line  272 in manuscript).

Manuscript: Material and methods, Line 272 (new)

3.7. Isolation of 4 phlorotannins from E.cava

E. cava used this study were obtained from Aqua Green Technology Co., Ltd (Jeju, Republic of Korea). In briefly, E. cava were thoroughly washed with pure water and air-dried at room temperature for 48 hrs, the clean E. cava was ground and 50% ethyl alcohol was added followed by incubation at 85℃ for 12 hrs. The extracts of E. cava were filtered, concentrated, sterilized by heating at high temperatures above 85 ℃ for 40-60 min and then spray-dried. Phlorotannins were isolated following a published  procedure [34]. In briefly, centrifugal partition chromatography was performed using a two phase solvent system full of water/ethyl acetate/methyl alcohol/n-hexane (7:7:3:2, ratio v/v/v/v). The organic stationary phase was filled in the centrifugal partition chromatography column followed by pumping of the mobile phase into the column in descending mode at the same flow rate (2 ml per min) used for separation.

Response 8-2: We appreciated this comment. As this comment, detailed descriptions of H/R (hypoxia/reoxygenation) treatment modeling are poor in manuscript. So, we added detailed description of H/R modeling in material and methods section of manuscript (Material and methods section, Line 191 in manuscript).

Manuscript: Material and methods, Line 191 (previous)

3.2. Kidney ischemia reperfusion cell models

To examine the inhibitory effects of 4 phlorotannins DK, PHB, PFFA and PPB from E. cava extract, we designed two types of ischemia/reperfusion cell models using TCMK-1 cells; the experimental scheme is shown in Figure S1.

3.2.1. Pre-ischemia model

Four prepared phlorotannins (2.5 μg/ml) were added to growth medium, which was used to treat TCMK-1 cells for 6 h before ischemia. After treatment, the medium was completely removed, and the cells were washed thoroughly with PBS. Ischemia medium contained 0.5% FBS and 1% penicillin/streptomycin in DMEM. The TCMK-1 cells were cultured in ischemia medium with the 4 phlorotannins and exposed to a mixture of 1% O2, 5% CO2, 94% N2 for 14 h. After incubation, the ischemia medium was removed, and the cells were washed with PBS. The cells were cultured in growth medium containing 4 phlorotannins in a 5% CO2 incubator for 9 h, and then used for experiments.

3.2.2. Post-ischemia model

TCMK-1 cells were cultured in growth medium for 6 h before ischemia, growth medium was completely removed, and the cells were washed with PBS. Ischemia treatment was performed as in 3.2.1. After the treatment, the ischemia medium was removed, and the cells were washed with PBS. The cells were cultured in growth medium containing 4 phlorotannins in a 5% CO2 incubator for 9 h, and then used for experiments.

Manuscript: Material and methods, Line 191 (new)

3.2. Hypoxia/reoxygenation (H/R) injury cell models

To examine the inhibitory effects of 4 phlorotannins DK, PHB, PFFA and PPB from E. cava extract, we designed two types of H/R cell models using TCMK-1 cells; the experimental method and image scheme is shown in Figure S1.

3.2.1. Pre-hypoxia treatment model

Four prepared phlorotannins (2.5 μg/ml) were added to growth medium, which was used to treat TCMK-1 cells for 6 h before hypoxia. After treatment, the medium was completely removed, and the cells were washed thoroughly with PBS. Hypoxia medium contained 0.5% FBS and 1% penicillin/streptomycin in DMEM. The TCMK-1 cells were cultured in hypoxia medium with the 4 phlorotannins and exposed to gas mixture of 1% O2, 5% CO2, 94% N2 for 14 h within hypoxia incubator chamber (Stemcell technologies; MA, USA). After flushing the hypoxia chamber with the desired gas mixture, the chamber seals tightly with clamp. After incubation, the hypoxia chamber was dismantled and the hypoxia medium was removed, and the cells were washed with PBS. The cells were cultured with fresh oxygenated medium containing FBS, 1% penicillin/streptomycin, 4 phlorotannins in a mixture of 20% O2, 5% CO2, 75% N2 for 9 h, and then used for experiments. The experimental method and image scheme is shown in Figure S1.

3.2.2. Post-hypoxia treatment model

TCMK-1 cells were cultured in growth medium for 6 h before hypoxia, growth medium was completely removed, and the cells were washed with PBS. Hypoxia exposure was performed as in 3.2.1. Pre-hypoxia treatment model of material and methods section. After the hypoxia treatment, the hypoxia chamber was dismantled and the hypoxia medium was removed, and the cells were washed with PBS. The cells were cultured with fresh oxygenated medium containing FBS, 1% penicillin/streptomycin, 4 phlorotannins in a mixture of 20% O2, 5% CO2, 75% N2 for 9 h, and then used for experiments. The experimental method and image scheme is shown in Figure S1.

Point 9: The results of MAPKs are not reported in the conclusion section.

Response 9: We appreciated this comment. As this comment, we added results of MAPKs (such as pErk, pSAPK/JNK, and pp38) in the conclusion section of manuscript (Conclusion section, Line 300).

Manuscript: Conclusion section, Line 265 (previous)

In conclusion, PPB showed a protective effect against I/R injury of TECs. PPB reduced HMGB1 release, which initiates TLR4 and RAGE activation. The decrease in TRL4 and RAGE signaling by PPB inhibited NF-κB activation, which leads to TNF-α and IL-6 release. Those cytokines are well-known inducers of TEC apoptosis after I/R injury. PPB reduced TNF-α and IL-6 release, and reduced cell death, which suggested that PPB has a potential to prevent kidney I/R injury by inhibiting the HMGB1-TLR4 or RAGE pathway. Among the 4 phlorotannins, PPB had the strongest protective effect from I/R injury, in both the pre- and post-ischemia models. Our data suggested that PPB treatment could be used to protect against I/R injury during kidney surgery before the blood supply to the kidney is stopped and even after ischemia, before the blood supply to the kidney is restored; the latter treatment might be as effective as when treatment with PPB is performed before the onset of ischemia.

Manuscript: Conclusion section, Line 300 (new)

In conclusion, PPB showed a protective effect against H/R injury of TECs. The PPB reduced HMGB1 release, which initiates TLR4 and RAGE activation. The decrease in TRL4 and RAGE signaling by PPB inhibited NF-κB activation, which leads to TNF-α and IL-6 release. Those cytokines are well-known inducers of TEC apoptosis after H/R injury. PPB reduced TNF-α and IL-6 release, and reduced cell death and cell death signaling molecules such as pErk1/2, pSAPK/JNK, and pP38, which suggested that PPB has a potential to prevent kidney H/R injury by inhibiting the HMGB1-TLR4 or RAGE pathway. Among the 4 phlorotannins, PPB had the strongest protective effect from H/R injury, in both the pre- and post-hypoxia models. Our data suggested that PPB treatment could be used to protect against I/R injury during kidney surgery before the blood supply to the kidney is stopped and even after ischemia, before the blood supply to the kidney is restored; the latter treatment might be as effective as when treatment with PPB is performed before the onset of ischemia.

Reviewer 3 Report

This is an interesting study, which shows that a polyphenol extracts (PPB) from the brown alga Ecklonia cava could attenuate tubular epithelia cell (TEC) death occurring in kidney surgery. This TEC death is diagnosticated as an ischemia/reperfusion (I/R) injury and therefore PPB could be a novel target for renal I/R injury therapy. TECs induce their own apoptosis due to the release from TECs of a protein called HMBG1. HMBG1 binds on Toll Like Receptor 4 (TLR4) triggering a TNF alpha and IL-6 dependent apoptosis. TNF and IL-6 genes activation are due to NF-kB and HMBG1 binds also on another receptor called RAGE inducing inflammation. Although the experiments carried out with Mouse renal tubular epithelial cells pre-ischemia and post-ischemia models are convincing to show that PPB decreases TEC death, a dose response experiment is necessary to demonstrate that PPB is blocking the HGMB1–TLR4/RAGE signaling pathway.

Minor points:

Line 82 : The most significant

Line 162: PPB attenuated cell death signals which induced by I/R injury

Author Response

Response to Reviewer 3 Comments

This is an interesting study, which shows that a polyphenol extracts (PPB) from the brown alga Ecklonia cava could attenuate tubular epithelia cell (TEC) death occurring in kidney surgery. This TEC death is diagnosticated as an ischemia/reperfusion (I/R) injury and therefore PPB could be a novel target for renal I/R injury therapy. TECs induce their own apoptosis due to the release from TECs of a protein called HMBG1. HMBG1 binds on Toll Like Receptor 4 (TLR4) triggering a TNF alpha and IL-6 dependent apoptosis. TNF and IL-6 genes activation are due to NF-kB and HMBG1 binds also on another receptor called RAGE inducing inflammation. Although the experiments carried out with Mouse renal tubular epithelial cells pre-ischemia and post-ischemia models are convincing to show that PPB decreases TEC death, a dose response experiment is necessary to demonstrate that PPB is blocking the HGMB1–TLR4/RAGE signalling pathway.

Point 1: (Minor points) Line 82 : The most significant

Response 1 : We appreciate this comment. We checked above sentence and changed ‘the significant’ to ‘the most significant’ in manuscript (Result and Discussion section in manuscript, Line 82). 

Manuscript: Result and Discussion section, Line 82 (previous)

In the post-ischemia model, the HMGB1 level increased by I/R injury was also decreased by adding E. cava extracts before reperfusion in both TEC lysate and supernatant. Among the 4 phlorotannins, the effect of DK and PPB was the significant (Fig. 1C, D).

Manuscript: Result and Discussion section, Line 82 (new)

In the post-ischemia model, the HMGB1 level increased by I/R injury was also decreased by adding E. cava extracts before reperfusion in both TEC lysate and supernatant. Among the 4 phlorotannins, the effect of DK and PPB was the most significant (Fig. 1C, D).

Point 2: (Minor points) Line 162: PPB attenuated cell death signals which induced by I/R injury

Response 2: We appreciate this comment. As this comment, we changed ‘PPB attenuated cell death signals which induced by I/R injury’ to ‘PPB attenuated cell death signals induced by I/R injury’ in manuscript (Result and Discussion section in manuscript, Line 162).

Manuscript: Result and Discussion section, Line 162 (previous)

2.5. PPB attenuated cell death signals which induced by I/R injury

Manuscript: Result and Discussion section, Line 162 (new)

2.5. PPB attenuated cell death signals induced by I/R injury

Round 2

Reviewer 1 Report

The authors have answered to the comments

Author Response

We appreciate your comments, the this research paper is even better.

Reviewer 2 Report

Figure S2 and Figure S3 should be moved from the supplementary section to the main manuscript.

Author Response

Response to Reviewer 2 Comments

Point 1: Figure S2 and Figure S3 should be moved from the supplementary section to the main manuscript.

Response 1: We appreciated this comment. As this comment, Figure S2 is added to main figure 3 and figure S3 is added to main figure 4. New figures, figure legends can find in line 140 and 157 of Manuscript and these related Results and discussion section are also changed (Figure and Figure legend section and Results and discussion section).

Reviewer 3 Report

I regret that a dose response experiment to demonstrate that PPB is blocking the HGMB1–TLR4/RAGE signalling pathway was not carried out as I requested.

Author Response

We apologize for this comment.

We did not check your previous important comments in a proper way.